# Line operators in 4d Chern-Simons theory and Cherkis bows

**Nafiz Ishtiaque[1] and Yehao Zhou[2]**

**1** IHES, 91440 Bures-sur-Yvette, France
**2** Kavli IPMU (WPI), UTIAS, The University of Tokyo, Kashiwa, Chiba 277-8583, Japan

## Abstract

We show that the phase spaces of a large family of line operators in 4d Chern-Simons theory with $GL_n$ gauge group are given by Cherkis bow varieties with $n$ crosses. These line operators are characterized by Hanany-Witten type brane constructions involving D3, D5, and NS5 branes in an $\Omega$-background. Linking numbers of the five-branes and mass parameters for the D3 brane theories determine the phase spaces and in special cases they correspond to vacuum moduli spaces of 3d $\mathcal{N} = 4$ quiver theories. Examples include line operators that conjecturally create T, Q, and L-operators in integrable spin chains.



# 1  Introduction

4d Chern-Simons (CS) theory [1,2] is a holomorphic-topological quantum field theory on a 4-manifold $\Sigma \times C$ where $\Sigma$ is a topological surface and $C$ is a Riemann surface with a holomorphic volume form $\omega$. It is a gauge theory with a complex gauge group $G_{\mathbb{C}}$. The only dynamical field in this theory is the Lie algebra valued connection $\mathcal{A}_x \mathrm{d}x + \mathcal{A}_y \mathrm{d}y + \mathcal{A}_{\bar{z}} \mathrm{d}\bar{z}$ where $x, y$ are local coordinates on $\Sigma$ and $z, \bar{z}$ are local holomorphic and anti-holomorphic coordinates on $C$. The action for the theory is:

$$\frac{1}{\hbar} \int_{\Sigma \times C} \omega \wedge \mathrm{tr}\left( \mathcal{A} \wedge (\mathrm{d} + \overline{\partial})\mathcal{A} + \frac{2}{3} \mathcal{A} \wedge \mathcal{A} \wedge \mathcal{A} \right). \tag{1.1}$$

The theory is topological on $\Sigma$ and holomorphic on $C$. Here we only focus on the case $\Sigma = \mathbb{R}^2$, $C = \mathbb{C}$, $\omega = \mathrm{d}z$ and the gauge group is $G_{\mathbb{C}} = \mathrm{GL}_n = \mathrm{GL}(n, \mathbb{C})$ with Lie algebra $\mathfrak{gl}_n = \mathfrak{gl}(n, \mathbb{C})$.

In this paper we study phase spaces of line operators in 4d CS theory that are local in the holomorphic direction $\mathbb{C}$. A line operator in a gauge theory with gauge group $G$ can be defined by taking a quantum mechanics with $H$-symmetry, a homomorphism $G \to H$ by which $G$ acts on its phase space, and coupling it to the gauge theory. By the phase space of the line operator we refer to the phase of this quantum mechanics. The line operators we study are motivated by certain brane constructions. We label these line operators in the $\mathrm{GL}_n$ theory by

$$
\begin{aligned}
\text{a spectral parameter,} \quad & z \\
\text{an } n\text{-tuple of integers,} \quad & \boldsymbol{K} = (K_1, \cdots, K_n) \\
\text{a } p\text{-tuple of integers,} \quad & \boldsymbol{L} = (L_1, \cdots, L_p) \\
\text{and, a } (p-1)\text{-tuple of complex numbers,} \quad & \boldsymbol{\varrho} = (\varrho_1^{\mathbb{C}}, \cdots, \varrho_{p-1}^{\mathbb{C}})
\end{aligned}
\tag{1.2}
$$

satisfying

$$N := \sum_{i=1}^{n} K_i = \sum_{i=1}^{p} L_i. \tag{1.3}$$

Our main result is that the line operator labeled by $z$, $\boldsymbol{\varrho}$, $\boldsymbol{K}$, and $\boldsymbol{L}$ – which we denote by $\mathbb{L}_{\boldsymbol{\varrho}}^z(\boldsymbol{K}, \boldsymbol{L})$ – can be described classically in terms of a complex symplectic space called a Cherkis bow variety, denoted in this paper by $\mathcal{M}_{\boldsymbol{\varrho}}^{\mathrm{bow}}(\boldsymbol{K}, \boldsymbol{L})$. Our result is similar to how flag varieties correspond to classical phase spaces of Wilson lines in 3d CS theory [3]. The bow varieties were introduced by Cherkis as certain moduli spaces of instantons [4–6] and further described by Nakajima and Takayama as quiver varieties [7] who also showed that for certain choices of $\boldsymbol{K}$ and $\boldsymbol{L}$ these varieties coincide with the Coulomb branches of 3d $\mathcal{N} = 4$ quiver gauge

theories. Note that the phase spaces are independent of the spectral parameter and we will omit references to this parameter for the most part.

To each line operator $\mathbb{L}_\varrho^z(K,L)$ we assign two 3d $\mathcal{N}=4$ theories which we denote by $T_\varrho[U(N)]_K^L$ and $T_\varrho^\vee[U(N)]_K^L$. These two theories are mirrors [8] of each other. The bow variety $\mathcal{M}_\varrho^{\text{bow}}(K,L)$ is the Coulomb branch of $T_\varrho[U(N)]_K^L$ and the Higgs branch of $T_\varrho^\vee[U(N)]_K^L$. When $K$ and $L$ satisfy certain constraints, the 3d theory $T_\varrho[U(N)]_K^L$ admits a linear quiver description. Coulomb branches of these quiver theories are known to be slices in the affine Grassmannian whose deformation quantization results in shifted truncated Yangians [9–12]. Line operators in 4d CS theory form integrable spin chains which carry natural Yangian actions [13,14]. Our construction then suggests that not only the (deformation) quantization of slices in the affine Grassmannian but also that of Bow varieties should result in algebras with RTT representations (e.g. shifted truncated Yangian). From this point of view, our results are closely related to the Bethe/Gauge correspondence of Nekrasov and Shatashvili [15–17], we will not discuss this connection in depth in this paper.

It was shown in [18] that bow varieties also arise as certain moduli spaces of vacua in 4d $\mathcal{N}=4$ theories with impurity walls. These defect 4d theories are effectively described by 3d $\mathcal{N}=4$ theories and the relevant solutions to the vacuum equations become the Higgs branches of these 3d theories. This establishes an equivalence between bow varieties and Higgs branches as hyper Kähler spaces. Using $\Omega$-deformation, we localize these defect 4d theories to 2d holomorphic BF theories with line defects. We then identify the complex phase spaces of these BF theories with the complex bow varieties described in [7]. Combinatorial data associated with Hanany-Witten brane configurations [19] involving D3-D5-NS5 branes were also used in the mathematical literature [20] to define bow varieties, which is precisely the association between bow varieties and Higgs branches of 3d $\mathcal{N}=4$ theories made in [18] and in our paper. These authors define stable envelopes for torus equivariant cohomology of bow varieties and being motivated by S-duality in string theory they study symplectic duality between stable envelopes of bow varieties defined by mirror Hanany-Witten configurations. The description of 3d $\mathcal{N}=4$ vacuum branches as bow varieties derived in [18] and this paper puts the results of [20] in a more physical context.

A similar characterization of some spins in terms of algebras appears in [21] in the context of studying solutions to the RTT relations in rational $\mathfrak{gl}_n$ spin chains. Given an R-matrix, a variety of T-operators, or more generally referred to as L-operators in [21] are characterized as follows. A module $V_q$ of $U(\mathfrak{gl}_q)\otimes\text{Weyl}^{\otimes q(n-q)}$, where Weyl is the Weyl algebra or the Heisenberg algebra with $q(n-q)$ oscillators, becomes an induced module for the Yangian $Y(\mathfrak{gl}_n)$ via a homomorphism $Y(\mathfrak{gl}_n)\to U(\mathfrak{gl}_q)\otimes\text{Weyl}^{\otimes q(n-q)}$. Given $V_q$, the authors of [21] construct solutions $L_q(z)\in\text{End}\left(\mathbb{C}^n\otimes V_q\otimes\mathbb{C}[z]\right)$ to the RTT relation that are linear in $z$. The operators $L_n$ and $L_1$ are called T and Q-operators respectively. [22] shows that the Q-operator can be computed from 4d CS theory by computing the expectation value of crossing Wilson and 't Hooft lines. Here Wilson and 't Hooft lines can be constructed as topological line defects labeled by the module $\mathbb{C}^n$ of $\mathfrak{gl}_n$ and the Fock module for $\text{Weyl}^{\otimes(n-1)}$ respectively. We shall show that the family of line operators that we study includes operators with phase spaces whose deformation quantization leads to the algebras $U(\mathfrak{gl}_q)\otimes\text{Weyl}^{\otimes q(n-q)}$ for all $q=1,\cdots,n$. It is therefore natural to conjecture that crossing these line operators with Wilson lines shall lead to the L-operators of [21], though we leave the computation of such expectation values for future work.

This paper is structured as follows. In §2 we discuss the brane construction of 4d CS theories with line operators. We follow the construction of 4d CS as the $\Omega$-deformed world-volume theory of a stack of D5 branes from [23]. A family of line operators in this theory can be constructed by introducing NS5 branes and suspending D3 branes between the five-branes. We label a line operator by the linking numbers of the five-branes – the linking numbers of the D5

and the NS5 branes constitute the tuples $K$ and $L$ respectively. The remaining parameters $z$ and $\varrho$ will denote locations of the NS5 branes in certain directions. Supersymmetric configurations of the D3 branes parameterize the classical phase spaces that we assign to the line operators created by the D3 branes. These supersymmetric configurations can be seen as the spaces of vacua of the world-volume theory of the D3 branes. In general we can describe this theory as a 4d $\mathcal{N} = 4$ theory on an interval, which in principle can effectively be described as a 3d $\mathcal{N} = 4$ theory at low energy. This defines the 3d theory $T_\varrho^\vee[\mathrm{U}(N)]_K^L$ and we define the theory $T_\varrho[\mathrm{U}(N)]_K^L$ as its mirror. The phase space assigned to the line operator is the Higgs branch of $T_\varrho^\vee[\mathrm{U}(N)]_K^L$ which is also the Coulomb branch of the mirror. For some restricted set of linking numbers we can rearrange the five and the three-branes using Hanany-Witten transitions in a way that leads to a standard description of the world-volume theory of the D3 branes as 3d $\mathcal{N} = 4$ quiver gauge theories [19]. In these quiver cases the $\varrho_i^\mathbb{C}$s will denote the complex Fayet–Iliopoulos (FI) parameters in $T_\varrho^\vee[\mathrm{U}(N)]_K^L$ and complex twisted masses in $T_\varrho[\mathrm{U}(N)]_K^L$. 3d $\mathcal{N} = 4$ theories admit FI parameters and twisted masses that are triplets of real scalars rotated by $\mathrm{SU}(2) \times \mathrm{SU}(2)$ R-symmetry acting on the hyper-Kähler vacuum branches [8]. However, we will land on a complex description of the vacuum branches as opposed to the hyper-Kähler description, and as such only a complex linear combination of two of the components of a real triplet will be prominent in our discussion.

In §3 and §4 we take a more field theoretic route to arrive at the description of the phase spaces as bow varieties. Instead of looking at the 3d $\mathcal{N} = 4$ descriptions, we look directly at the 4d world-volume theory of the D3 branes with boundaries and domain walls provided by the five-branes. The 4d theory is $\mathcal{N} = 4$ super Yang-Mills (SYM) and the five-branes impose 1/2-BPS boundary conditions on the fields of this theory [24, 25]. In §3 we check that these 1/2-BPS boundaries preserve the Kapustin-Witten twist [26] of the 4d $\mathcal{N} = 4$ SYM labeled by the twist parameter $t = \mathrm{i}$ (as defined in §3.1 of [26], see also (3.26)). In §4 we show that turning on $\Omega$-deformation reduces this twisted 4d theory to 2d BF theory on an interval. In this setup the phase space attached to the line operator is the phase space of the BF theory, which is the moduli space of solutions to complex Nahm's equation with boundary conditions. We determine the boundary conditions in the BF theory descended from boundary conditions of the original 4d theory and we show that the phase spaces coincide with Cherkis bow varieties $\mathcal{M}_\varrho^{\mathrm{bow}}(K, L)$. When the linking numbers are restricted such that $T_\varrho[\mathrm{U}(N)]_K^L$ admits a quiver description we check that $\mathcal{M}_\varrho^{\mathrm{bow}}(K, L)$ is indeed the coulomb branch of $T_\varrho[\mathrm{U}(N)]_K^L$ using results from [7]. In the final section §5 we discuss some examples of line operators, their phase spaces, and the quantized algebras. As special cases we find line operators whose quantization give algebras related to the T, Q, and L-operators as described in [21].

The main contributions of the paper are as follows. We show that Bow varieties provide a geometric characterization of spins in rational $\mathfrak{gl}_n$ spin chains. We do this by showing that these varieties are phase spaces of line operators in 4d CS theory and then appealing to the known relation between these line operators and integrable spin chains. The result suggests that bow varieties can be endowed with the structure of classical integrable systems which quantize to rational $\mathfrak{gl}_n$ spin chains. We identify these phase spaces with supersymmetric vacua of 4d $\mathcal{N} = 4$ SYM theories on intervals with domain walls, as in [18]. These vacua correspond to the Higgs branch vacua of some effective 3d theories, or equivalently, to the Coulomb branch vacua of the mirror theories. Applying $\Omega$-deformation to the 4d setup we show that a protected subsector of these 4d theories can be described as 2d BF theories with boundaries and line defects. We map boundary conditions and domain walls of the 4d theory to those of the BF theory. The bow varieties become the complex phase spaces of these defect BF theories.

## 2 Brane Construction of Line Operators

### 2.1 General Considerations (Arbitrary Linking Numbers)

4d CS theory with $GL_n$ gauge group can be constructed from a stack of $n$ D5 branes. Let $\Sigma$ be a a 2d surface, $C$ a Riemann surface, and TN the 4d Taub-NUT space. We start in type IIB string theory with the 10d space-time being $T^*\Sigma \times C \times TN$. The TN can be described very concretely in terms of a coordinate $\vec{x}$ of $\mathbb{R}^3$ and a circle with coordinate $\theta$ – in terms of which the TN metric can be written as:

$$\mathrm{d}s^2_{\mathrm{TN}} = U \mathrm{d}\vec{x} \cdot \mathrm{d}\vec{x} + \frac{1}{U}(\mathrm{d}\theta + \vec{\omega} \cdot \mathrm{d}\vec{x})^2\,, \tag{2.1}$$

where $U = \frac{1}{r} + \frac{1}{\lambda^2}$ and $\vec{\omega}$ is a vector field on $\mathbb{R}^3\backslash\mathbb{R}$ satisfying $\mathrm{d}U = \star_{\mathbb{R}^3}\mathrm{d}(\vec{\omega} \cdot \mathrm{d}\vec{x})$. We see that the TN circle collapses at the center of $\mathbb{R}^3$ and at infinity its radius asymptotes to $\lambda$. $\mathbb{R}^3$ can be parameterized by a radial coordinate $\rho := \sqrt{\vec{x} \cdot \vec{x}}$ and two angular coordinates. A 2d surface inside TN located at fixed values of these two angular coordinates and parameterized by $r$ and $\theta$ has the shape of a "cigar". TN can therefore be described as a family of cigars parameterized by an $S^2$, all the cigars sharing a single point – the tip – located at the center of the TN. We need to fix a specific supergravity background, the defining characteristic of the background is that it preserves a supercharge $Q$ which induces a B-type $\Omega$-deformation of the world-volume theory of any brane wrapping a cigar in TN. This requires turning on – in addition to a nontrivial metric – a dilaton and a RR 2-form. For details about this particular background we refer to [23].[1] Here we simply note that if any D-brane is placed in this background wrapping a cigar inside TN then from the point of view of the cigar the D-brane theory looks like an $\Omega$-deformed B-model. The supercharge $Q$ squares to a rotation of the TN circle which we schematically write as $Q^2 \sim \mathcal{L}_{\hbar\partial_\theta}$, here $\partial_\theta$ is the vector field generating rotation of the TN circle and $\hbar$ acts as a deformation parameter.

To find 4d CS theory we introduce a stack of $n$ D5 branes wrapping $\Sigma \times C \times \mathrm{Cig}$ where Cig is some chosen cigar inside TN. The world-volume theory of the D5 branes is 6d $\mathcal{N} = (1,1)$ $U(n)$ SYM, which upon $\Omega$-deformation along Cig reduces to 4d $GL_n$ CS theory on $\Sigma \times C$ at the level of BRST cohomology [23]. To get line operators in this theory we introduce NS5 branes that share 3 directions with the D5s, wrap the entire TN, and do not wrap any direction in $C$. We further introduce D3 branes suspended between the D5s and the NS5s. The D3 branes have finite extent in one of the directions and at low energy the corresponding world-volume theory is a 3d $\mathcal{N} = 4$ theory, which upon $\Omega$-deformation becomes a 1d topological quantum mechanics[2] (TQM) coupled to the 4d CS theory [28,29]. Different configurations of D3 and NS5 branes lead to different TQMs and each TQM defines a line operator in the 4d CS theory.

We shall simplify our discussions slightly by taking $\Sigma = \mathbb{R}^2$ and $C = \mathbb{C}$. For notational simplicity we also write TN as $\mathbb{R}^2_{+\hbar} \times \mathbb{R}^2_{-\hbar}$. The notation is meant to imply that the supercharge $Q$ squares to a bosonic symmetry that rotates $\mathbb{R}^2_{+\hbar}$ and $\mathbb{R}^2_{-\hbar}$ in opposite angular directions. These two planes can be thought of as two antipodal cigars inside TN. We choose coordinates with indices running from 0 to 9 to label directions in our 10d space-time and we summarize the directions wrapped by the D3, D5, and NS5 branes in Table 1.

The NS5 branes and consequently the D3 branes have fixed positions in the $C$ direction. To create a single line operator these branes must be coincident in the $C$ direction, the location of these branes in this direction will be the spectral parameter associated with the line operator. Different numbers of NS5s and D3s lead to different line operators. Since we want to create 4d CS theory with $GL_n$ gauge group, the number of D5 branes is fixed to be $n$. Suppose

---

[1]Some T-dual version of this background was introduced in [27] with the name Taub-trap background.

[2]Topological means that the action for the quantum mechanics depends only on the symplectic form of the target, there is no Hamiltonian.

Table 1: Directions wrapped by various branes in the construction of 4d CS theory with line operators. The 10d space-time is $T^*\Sigma \times C \times TN$. We have replaced TN with $\mathbb{R}^2_{+\hbar} \times \mathbb{R}^2_{-\hbar}$ and we have chosen indices for 10d coordinates parameterizing various components as follows: $\Sigma = \mathbb{R}^2_{07}$, $T^*\Sigma = \mathbb{R}^4_{0734}$, $C = \mathbb{C} = \mathbb{R}^2_{89}$, $\mathbb{R}^2_{+\hbar} = \mathbb{R}^2_{12}$, and $\mathbb{R}^2_{-\hbar} = \mathbb{R}^2_{56}$. We write $\mathbb{R}^n_{i_1 \cdots i_n}$ to refer to the $\mathbb{R}^n$ space parameterized by coordinates $x^{i_1}, \cdots, x^{i_n}$.

|     |   | $\mathbb{R}^2_{+\hbar}$ |   |   |   | $\mathbb{R}^2_{-\hbar}$ |   |   | $C$ |   |
| --- | - | - | - | - | - | - | - | - | - | - |
|     | 0 | 1 | 2 | 3 | 4 | 5 | 6 | 7 | 8 | 9 |
| D5  | × | × | × |   |   |   |   | × | × | × |
| D3  | × | × | × | × |   |   |   |   |   |   |
| NS5 | × | × | × |   | × | × | × |   |   |   |

there are $p$ NS5 branes. We choose an ordering of the five-branes in the $x^3$ direction and reading from left to right, we label the NS5 branes as $\mathrm{NS5}_1, \ldots, \mathrm{NS5}_p$ and the D5 branes as $\mathrm{D5}_1, \ldots, \mathrm{D5}_n$. A particular configuration can now be described by an $n$-tuple $\boldsymbol{K} = (K_1, \cdots, K_n)$ and a $p$-tuple $\boldsymbol{L} = (L_1, \cdots, L_p)$ of integers satisfying $\sum_{i=1}^n K_i = \sum_{i=1}^p L_p$ where $K_i$ and $L_i$ are the linking numbers of $\mathrm{D5}_i$ and $\mathrm{NS5}_i$ respectively (concept introduced in [19] but our terminology is from [25]):[3]

$$K_i := \text{No. of (D3s to the left – D3s to the right + NS5s to the right) of } \mathrm{D5}_i$$
$$L_i := \text{No. of (D3s to the right – D3s to the left + D5s to the left) of } \mathrm{NS5}_i \tag{2.2}$$

Here left and right refers to the $x^3$ direction. If we put all the NS5 branes to the left of all the D5 branes then the linking numbers are simply the numbers of D3 branes ending on the D5 (resp. NS5) branes from the left (resp. right). We depict the corresponding configuration pictorially in Fig. 1. We only impose the constraints on the linking numbers that our diagrams do not

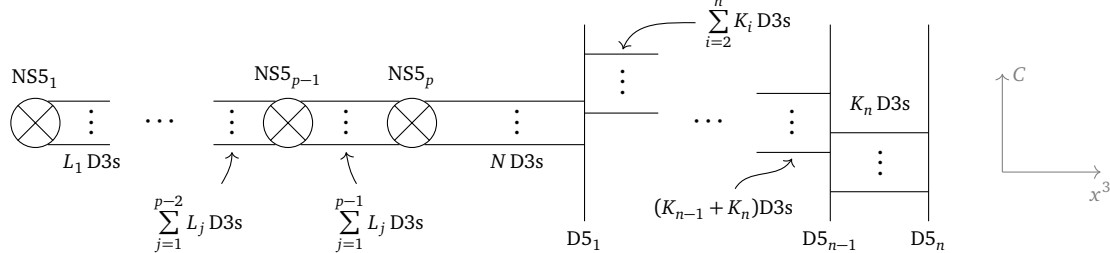

Figure 1: Brane configuration for the line operator $\mathbb{L}_\varrho(\boldsymbol{K}, \boldsymbol{L})$ in $\mathrm{GL}_n$ 4d CS theory labeled by linking numbers $\boldsymbol{K} = (K_1, \cdots, K_n)$ and $\boldsymbol{L} = (L_1, \cdots, L_p)$ satisfying $\sum_{i=1}^n K_i = \sum_{j=1}^p L_j = N$. $\hbar\varrho = \hbar(\varrho_1^{\mathbb{C}}, \cdots, \varrho_{p-1}^{\mathbb{C}})$ are complex FI parameters, they are not visible in the classical brane picture. We denote the 3d $\mathcal{N} = 4$ theory describing the low energy dynamics of the D3 branes by $T_\varrho^\vee[\mathrm{U}(N)]_K^L$.

become disconnected, i.e., there are nonzero number of D3 branes between any two adjacent five-branes and that the brane configurations remain supersymmetric, i.e., the s-rule [19] is

---

[3]To be precise, $L_i$ – what we are calling the linking number of the $i$th NS5 brane – is really $n$ minus the linking number of the NS5 brane where $n$ is the total number of D5 branes. However we shall keep referring to $L_i$ as the linking number of $\mathrm{NS5}_i$ simply for convenience. Our notion of linking number coincides precisely with that of the "charge of a five-brane" from [20].

not violated. These require $\sum_{j=i}^{n} K_j > 0$ for all $1 \le i \le n$, $\sum_{i=1}^{j} L_i > 0$ for all $1 \le j \le p$, and $N \le np$ for example, though these conditions are not sufficient in general.

The D3 branes have finite length in the $x^3$ direction and at low energy their world-volume theory is a 3d $\mathcal{N} = 4$ theory on $\mathbb{R}_{012}^3$. We denote the 3d $\mathcal{N} = 4$ theory describing the D3 branes at low energy by $T^\vee[\mathrm{U}(N)]_K^L$. This theory can be deformed by both FI parameters and twisted masses. Turning on FI parameters deforms the Higgs branch and turning on generic twisted masses lifts most of the Higgs branch leaving only isolated vacua [30–32]. We only turn on FI parameters, keeping the twisted masses zero, so that we have smooth Higgs branches. In terms of the branes, FI parameters correspond to differences in locations of the NS5 branes in the $\mathbb{R}_{789}^3$ direction [19]. We have already mentioned that we take the NS5 branes to be coincident in these directions to have a single line operator. In practice we will assume that the differences in their locations in these direction are of order $\mathcal{O}(\hbar)$ so that classically they are still coincident but these differences will generate FI deformations of order $\mathcal{O}(\hbar)$ in the theory. Thus the 3d theory in general can be deformed by turning on the FI parameters $\hbar \vec{\varrho}_j = \hbar(\varrho_{1,j}, \varrho_{2,j}, \varrho_{3,j})$ where $\hbar \vec{\varrho}_j$ can schematically be interpreted as the difference between the locations of NS5$_j$ and NS5$_{j+1}$ in the $\mathbb{R}_{789}^3$ direction. In the $\Omega$-background we get a holomorphic description of the Higgs branch which is deformed by the parameters associated with the locations of the NS5 branes in the $C$ direction. We therefore define the complex FI parameters

$$\hbar \varrho_j^{\mathbb{C}} := \hbar(\varrho_{j,2} - \mathrm{i}\varrho_{j,3}). \tag{2.3}$$

The remaining FI parameter, also called the real FI parameter, is traded for stability conditions during the construction of Higgs branches as invariant quotients [33]. We therefore label our theory using only the complex parameters: $T_\varrho^\vee[\mathrm{U}(N)]_K^L$ where $\varrho := (\varrho_1^{\mathbb{C}}, \cdots, \varrho_{p-1}^{\mathbb{C}})$. For arbitrary linking numbers $K$ and $L$ a more concrete description of this theory is not immediate. To create a line operator we do not assume any constraints on the linking numbers other than preservation of supersymmetry.

We shall denote the Higgs branch of a theory $T$ by $\mathcal{M}_H(T)$. A 3d $\mathcal{N} = 4$ theory on $\mathbb{R}_{012}^3$ with B-type $\Omega$-deformation on the $\mathbb{R}_{12}^2$ plane localizes to a TQM on $\mathbb{R}_0$ whose target is the Higgs branch of the 3d theory [28, 29]:[4]

$$T \text{ on } \mathbb{R}_{012}^3 \xrightarrow{\Omega\text{-deformation on } \mathbb{R}_{12}^2} \text{TQM on } \mathbb{R}_0 \text{ with the target } \mathcal{M}_H(T). \tag{2.4}$$

Before the localization by $\Omega$-background, the 3d theory couples to the 6d $\mathcal{N} = (1,1)$ U$(n)$ SYM theory of the D5 branes as a 3d defect. After localization we find the aforementioned TQM coupled to the 4d GL$_n$ CS theory. The 4d CS connection couples to the flavor current of the TQM. The flavor symmetry of the TQM is the complexification of the flavor symmetry of the Higgs branch of the 3d $\mathcal{N} = 4$ theory, which depends on the exact brane configuration and is difficult to describe concisely in general. Let us schematically denote this flavor symmetry of the 3d theory by $F(\boldsymbol{K}, \boldsymbol{L})$:

$$F(\boldsymbol{K}, \boldsymbol{L}) := \text{Higgs branch flavor symmetry group of } T_\varrho^\vee[\mathrm{U}(N)]_K^L, \tag{2.5}$$

and the flavor symmetry of the TQM by $F_{\mathbb{C}}(\boldsymbol{K}, \boldsymbol{L})$. The flavor symmetry of the 3d theory is a subgroup of the U$(n)$ that rotates the $n$ D5 branes. After complexification by $\Omega$-background we therefore find $F_{\mathbb{C}}$ to be a subgroup of GL$_n$ and the $\mathfrak{gl}_n$-valued connection $\mathcal{A}$ of the 4d CS theory couples to the TQM via $\int_{\mathbb{R}} \mathrm{tr}(\mathcal{A} \mu_{\mathcal{M}_H})$ where $\mu_{\mathcal{M}_H}$ is the (dual of the) Lie$(F_{\mathbb{C}})$ valued

---

[4]This quantum mechanics can also be studied as a protected sub-sector of the 3d $\mathcal{N} = 4$ superconformal algebra [34–37]. The relation between certain protected sub-sectors of a superconformal algebras and $\Omega$-deformations persists in 4d $\mathcal{N} = 2$ (and possibly more generally) and in fact the 3d version can be derived as a dimensional reduction thereof [38, 39].

moment map on $\mathcal{M}_H$. This TQM defines a line operator in the 4d CS theory which we denote by $\mathbb{L}_\varrho(K, L)$. The phase space of the line operator is the target of the TQM:

$$\text{Phase space of } \mathbb{L}_\varrho(K, L) = \mathcal{M}_H(T_\varrho^\vee[\mathrm{U}(N)]_K^L), \tag{2.6}$$

and the TQM quantizes (the holomorphic functions on) this phase space into an operator algebra which we denote by $\mathcal{A}_\varrho(K, L)$:

$$\mathcal{A}_\varrho(K, L) := \text{Operator algebra that couples to } \mathbb{L}_\varrho(K, L). \tag{2.7}$$

## 2.2 Special Cases (with Quiver Descriptions)

In the remainder of this section, let us look at the cases where the 3d theory $T_\varrho^\vee[\mathrm{U}(N)]_K^L$ admits a quiver description as examples.[5]

In order to have a quiver description, we impose the following constraints on the linking numbers:

$$0 < K_i < p, \qquad 1 \leq i \leq n,$$

$$\widetilde{v}_j := \sum_{i=1}^{j} L_i - \sum_{i=1}^{j-1} (j-i)\widetilde{w}_i \geq 0, \qquad 0 < j < p, \tag{2.8}$$

$$\text{where,} \quad \widetilde{w}_j := \#\{K_i \,|\, K_i = p - j\}.$$

With these constraints we can bring the D5 branes between the NS5 branes using Hanany-Witten transitions [19] such that there are equal number of D3 branes on both sides of any D5 brane. The brane configuration after the Hanany-Witten transitions can be depicted as in Fig. 2. By standard arguments [19], the configuration of Fig. 2 leads to the 3d gauge theory

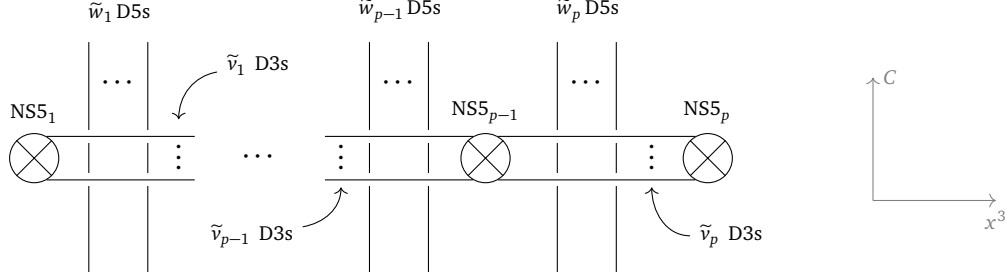

Figure 2: A brane configuration found after applying some Hanany-Witten transitions to the configuration in Fig. 1, assuming the constraints (2.8). The two brane configurations lead to the same IR description of the D3 brane world-volume theory in terms of the same 3d $\mathcal{N} = 4$ gauge theory.

defined by the quiver in Fig. 3. This 3d theory lives on $\mathbb{R}^3_{012}$ with B-type $\Omega$-deformation turned on with respect to the rotation of the $\mathbb{R}^2_{12}$ plane. In other words, this theory can be viewed as an $\Omega$-deformed 2d B-model on $\mathbb{R}^2_{12}$ with matter fields valued in the infinite dimensional hyper-Kähler target space $\mathrm{Map}(\mathbb{R}_0, X)$ where

$$X := \bigoplus_{i=1}^{p-1} T^*\mathrm{Hom}(\mathbb{C}^{\widetilde{v}_i}, \mathbb{C}^{\widetilde{w}_i}) \oplus \bigoplus_{i=1}^{p-2} T^*\mathrm{Hom}(\mathbb{C}^{\widetilde{v}_i}, \mathbb{C}^{\widetilde{v}_{i+1}}), \tag{2.9}$$

---

[5]For comparison, the Higgs branches of these theories would be bow varieties given by what [7] calls cobalanced dimension vectors.

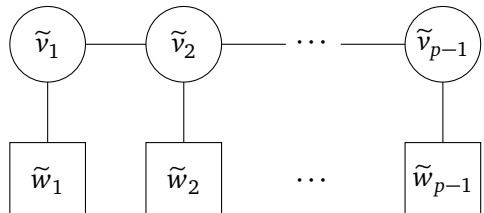

Figure 3: Quiver for the 3d $\mathcal{N} = 4$ theory $T_{\varrho}^{\vee}[\mathrm{U}(N)]_K^L$ with $N = \sum_{i=1}^n K_i = \sum_{j=1}^p L_j$. The ranks of the gauge and flavor groups are defined form $K$ and $L$ via (2.8). The complex FI parameter associated to the abelian factor of the $j$th gauge node is $\hbar \varrho_j^{\mathbb{C}}$ and $\boldsymbol{\varrho} = (\varrho_1^{\mathbb{C}}, \cdots, \varrho_{p-1}^{\mathbb{C}})$.

and the gauge group $\mathrm{Map}(\mathbb{R}_0, G)$ where

$$G := \prod_{i=1}^{p-1} \mathrm{U}(\widetilde{v}_i). \tag{2.10}$$

The $\Omega$-deformation localizes the 2d theory to a 1d TQM supported on $\mathbb{R}_0$ with the target being the Higgs branch of the 3d theory, as a complex symplectic manifold. Note that the Higgs and Coulomb branches of 3d $\mathcal{N} = 4$ theories are naturally hyper-Kähler spaces, but the $\Omega$-background breaks the SU(2) symmetry rotating the complex and symplectic structures down to a U(1) fixing some particular structures. The Higgs branch as a complex manifold is simply the symplectic reduction of $X$ by the complexified gauge group $G_{\mathbb{C}} = \prod_{i=1}^{p-1} \mathrm{GL}_{\widetilde{v}_i}$, subject to stability conditions:

$$\mathcal{M}_H(T_{\varrho}^{\vee}[\mathrm{U}(N)]_K^L) = X /\!\!/_{\varrho} G_{\mathbb{C}}. \tag{2.11}$$

This TQM couples to the 4d CS theory and creates the line operator $\mathbb{L}_{\varrho}(K, L)$. The Higgs branch flavor symmetry in the 3d theory is $F := \prod_{i=1}^{p-1} \mathrm{SU}(\widetilde{w}_i)$ which complexifies after the $\Omega$-deformation to the flavor symmetry $F_{\mathbb{C}} = \prod_{i=1}^{p-1} \mathrm{SL}_{\widetilde{w}_i}$ of the TQM. This TQM defines the line operator in the 4d CS theory whose phase space is the Higgs branch (2.11).

# 3 Boundary Conditions and Topological Twists in 4d $\mathcal{N} = 4$ SYM

For more general linking numbers we will not have nice quiver descriptions of the 3d theories associated to line operators as we did in the special cases of §2.2. Instead, we will analyze the D3 brane world-volume theory as a 4d $\mathcal{N} = 4$ theory on a finite interval with boundaries and domain walls. Moreover, we need to look at the $\Omega$-deformation of this theory. In this section we study the supersymmetry preserved by boundaries and domain walls to identify the topological twist of the 4d theory that will be relevant in the context of $\Omega$-deformation.

Type IIB string theory in flat $\mathbb{R}^{1,9}$ background has 32 real supercharges parameterized by two chiral spinors of the same chirality. We can take them to be in the **16** representation of SO(1,9). A stack of D3 branes breaks half of the supersymmetry by relating the two chiral spinors. Let us denote the remaining independent spinor by $\varepsilon$. The chirality constraint on $\varepsilon$ is:

$$\Gamma_{0\cdots9}\varepsilon = \varepsilon, \tag{3.1}$$

where $\Gamma_I$ for $I \in \{0, \cdots, 9\}$ are the SO(1,9) gamma matrices and $\Gamma_{I_1 \cdots I_k}$ refers to the completely anti-symmetrized product $\Gamma_{I_1 \cdots I_k} := \frac{1}{k!} \sum_{\sigma \in \mathfrak{S}_k} (-1)^{\mathrm{sgn}(\sigma)} \Gamma_{I_{\sigma(1)}} \cdots \Gamma_{I_{\sigma(k)}}$ where the sum is

over the symmetric group of order $k!$ and $\mathrm{sgn}(\sigma)$ is the signature of the permutation $\sigma$. The world-volume theory of the D3 branes is the 4d SYM with $\mathcal{N} = 4$ supersymmetry which preserves 16 supercharges. The D3-D5, as well as the D3-NS5, boundary preserves half of these supercharges. In fact, when placed as in Table 1, both boundaries preserve exactly the same 8 supercharges [24]. We briefly recap the boundary conditions on the fields of the 4d $\mathcal{N} = 4$ SYM at the two boundaries, following [24].

## 3.1  1/2-BPS Boundary Conditions in 4d $\mathcal{N} = 4$ SYM

Bosonic fields of the 4d $\mathcal{N} = 4$ SYM theory with a compact gauge group $G$ consist of a connection $A_0 dx^0 + \cdots + A_3 dx^3$ and six adjoint scalars $\phi_4, \cdots, \phi_9$ which correspond to fluctuations of the D3 branes in the transverse $\mathbb{R}^6_{4\ldots9}$ space. The bosonic symmetry of the theory is $SO(1,3) \times SO(6)$ where $SO(1,3)$ is the isometry of the space-time and $SO(6)$ rotates the six scalars. A 1/2-BPS boundary containing the temporal direction breaks this symmetry down to

$$I := SO(1,2) \times SO(3) \times SO(3), \tag{3.2}$$

where $SO(1,2)$ is the isometry of the 3d boundary and the two factors of $SO(3)$ each rotates three of the six scalars.[6] Looking at the brane configuration of Table 1 it is apparent that the D3-D5-NS5 configuration preserves the isometry of $\mathbb{R}^{1,2}_{012}$, an $SO(3)$ R-symmetry rotating $\mathbb{R}^3_{456}$, and another $SO(3)$ R-symmetry rotating $\mathbb{R}^3_{789}$. According to this symmetry breaking, we rename the six scalars as:

$$\vec{X} = (X_1, X_2, X_3) := (\phi_7, \phi_8, \phi_9), \qquad \vec{Y} = (Y_1, Y_2, Y_3) := (\phi_4, \phi_5, \phi_6), \tag{3.3}$$

and label the respective R-symmetries by $SO(3)_X$ and $SO(3)_Y$. Under the symmetry $SO(1,3) \times SO(6)$ of the 4d theory without boundaries, the spinors parameterizing the supersymmetry transform as:

$$S_\ell \oplus S_r := (\mathbf{2}, \mathbf{1}, \overline{\mathbf{4}}) \oplus (\mathbf{1}, \mathbf{2}, \mathbf{4}). \tag{3.4}$$

Here $(\mathbf{2}, \mathbf{1})$ and $(\mathbf{1}, \mathbf{2})$ refer to the left and the right handed representations of $SO(1,3)$ whereas $\mathbf{4}$ and $\overline{\mathbf{4}}$ refer to the two four dimensional spinor representations of $SO(6)$. Under the $SO(1,2) \subseteq I$ isometry of the boundary both left and right handed spinors of $SO(1,3)$ transform as $\mathbf{2}$. Under the remaining R-symmetry $SO(3) \times SO(3) \subseteq I$, both $\mathbf{4}$ and $\overline{\mathbf{4}}$ transform as $(\mathbf{2}, \mathbf{2})$. Therefore, under the symmetry $I$ preserved by the boundaries the two representations $S_\ell$ and $S_r$ become isomorphic:

$$S_\ell = S_r = V_8 := (\mathbf{2}, \mathbf{2}, \mathbf{2}). \tag{3.5}$$

As a representation of $I$ we thus have $S_\ell \oplus S_r = V_8 \otimes V_2$ where $V_2 = \mathbb{R}^2$. A key result from [24] is that for any choice of $\varepsilon_0 \in V_2$, there is a 1/2-BPS boundary condition of 4d $\mathcal{N} = 4$ SYM preserving all the supercharges parameterized by spinors of the form $v \otimes \varepsilon_0 \in V_8 \otimes V_2$.

There is an $SL(2, \mathbb{R})$ group acting on $V_2$ generated by the even elements of the 10d Clifford algebra that commute with $I$, namely:

$$B_0 = \Gamma_{456789}, \qquad B_1 = \Gamma_{3456}, \qquad B_2 = \Gamma_{3789}. \tag{3.6}$$

They satisfy $-B_0^2 = B_1^2 = B_2^2 = 1$, $B_0 B_1 = -B_1 B_0 = B_2$, etc. On $V_2$ these can be represented as:

$$B_0 = \begin{pmatrix} 0 & 1 \\ -1 & 0 \end{pmatrix}, \qquad B_1 = \begin{pmatrix} 0 & 1 \\ 1 & 0 \end{pmatrix}, \qquad B_2 = \begin{pmatrix} 1 & 0 \\ 0 & -1 \end{pmatrix}. \tag{3.7}$$

---

[6]The fermions transform under spin representations of $I$.

In order for a symmetry to be preserved by a boundary, the normal component of the associated current must vanish at that boundary. We shall impose this constraint on the supersymmetry generated by a spinor $u \otimes \varepsilon_0$. The 4d $\mathcal{N} = 4$ vector multiplet contains fermions in the same spinor representation as the spinors parameterizing the supersymmetry. Therefore, at the boundary the fermion is valued in $V_8 \otimes V_2$, just like the supersymmetry generators. Assuming that at the boundary the fermion takes the form $v \otimes \vartheta$, [24] spells out the boundary conditions for the vector multiplet fields needed to preserve the supersymmetry generated by spinors of the form $u \otimes \varepsilon_0$:[7]

$$\overline{\varepsilon}_0 (F_{\mu\nu} - \epsilon_{\mu\nu\lambda} F^{3\lambda} B_0) \vartheta = 0, \tag{3.8a}$$

$$D_\mu X_a (\overline{\varepsilon}_0 B_1 \vartheta) = 0, \tag{3.8b}$$

$$D_\mu Y_a (\overline{\varepsilon}_0 B_2 \vartheta) = 0, \tag{3.8c}$$

$$[X_a, Y_b](\overline{\varepsilon}_0 B_0 \vartheta) = 0, \tag{3.8d}$$

$$\overline{\varepsilon}_0 ([X_b, X_c] - \epsilon_{abc} D_3 X_a B_2) \vartheta = 0, \tag{3.8e}$$

$$\overline{\varepsilon}_0 ([Y_a, Y_b] - \epsilon_{abc} D_3 Y_c B_1) \vartheta = 0. \tag{3.8f}$$

Here $\mu, \nu, \dots \in \{0, 1, 2\}$ and $a, b, \dots \in \{1, 2, 3\}$. $\overline{\varepsilon}_0$ is the row vector $\begin{pmatrix} t & -s \end{pmatrix}$ for any column vector $\varepsilon_0 = \begin{pmatrix} s \\ t \end{pmatrix}$. $F$ is the curvature of $A$. Let us now focus separately on the D5 and the NS5 boundaries.

### 3.1.1 D5 Boundary

At the D3-D5 boundary the $\vec{Y}$ fields will satisfy Dirichlet boundary conditions since the D5 branes have fixed locations in the $\mathbb{R}^3_{456}$ space. For now, let us set the constant value of the $\vec{Y}$ fields to zero:

$$\vec{Y}|_{\text{D5}} = 0. \tag{3.9}$$

The constant value depends on the locations of the D5 branes in the $\mathbb{R}^3_{456}$ directions. Here we are not being specific about the number of D5 branes and we assume all the D5 branes to be located at the origin. A few comments about these locations are made in Remark 4.1.

(3.9) satisfies (3.8c) and (3.8d). The remaining equations from (3.8) are satisfied if we impose the following constraints on $A_\mu$ and $X$ at the D5 boundary:

$$\epsilon_{\lambda\mu\nu} F^{3\lambda} + \gamma F_{\mu\nu} = 0, \tag{3.10a}$$

$$D_3 X_a + \frac{u}{2} \epsilon_{abc} [X_b, X_c] = 0, \tag{3.10b}$$

for some numbers $\gamma$ and $u$ along with the following constraints on the vectors $\varepsilon_0$ and $\vartheta$:

$$\overline{\varepsilon}_0 (1 + \gamma B_0) \vartheta = 0, \tag{3.11a}$$

$$\overline{\varepsilon}_0 B_1 \vartheta = 0, \tag{3.11b}$$

$$\overline{\varepsilon}_0 (1 + u B_2) \vartheta = 0. \tag{3.11c}$$

By an overall scaling, which leaves (3.11) invariant, we can choose $\varepsilon_0$ to be of the form:

$$\varepsilon_0 = \begin{pmatrix} a \\ 1 \end{pmatrix}, \qquad a \in \mathbb{R} \cup \{\infty\}. \tag{3.12}$$

---

[7]Note that compared to the equations presented in §2.1 of [24] our equations have $B_1$ and $B_2$ swapped. The reason is that, we use the same definitions for $B_1$ and $B_2$ as in [24] and just like [24] we use $\vec{X}$ and $\vec{Y}$ to refer to coordinates parallel and normal to D5 branes, but our D5 branes wrap $\mathbb{R}^3_{789}$ whereas the D5 branes in [24] wrap $\mathbb{R}^3_{456}$.

The equations (3.11) now completely fix $\gamma$, $u$, and $\vartheta$ (up to scaling) in terms of $a$:

$$\gamma = \frac{a^2 - 1}{2a}, \qquad u = \frac{a^2 - 1}{a^2 + 1}, \qquad \vartheta = \begin{pmatrix} 1 \\ a \end{pmatrix}. \tag{3.13}$$

Thus, for any value of $a$ there is a set of D5-type 1/2-BPS boundary conditions on the bosonic fields given by (3.9) and (3.10) where $\gamma$ and $u$ are determined by (3.13). The standard Dirichlet boundary condition on the gauge field sets the components of the curvature parallel to the boundary to zero at the boundary. We achieve this by choosing:

$$a = \infty. \tag{3.14}$$

In this limit $\gamma \to \infty$ and $u \to 1$, turning (3.10) into the ordinary Dirichlet condition for the gauge field and Nahm's equation for $\vec{X}$:

$$F_{\mu\nu}|_{\text{D5}} = 0, \tag{3.15a}$$

$$\left( D_3 X_a + \frac{1}{2}\epsilon_{abc}[X_b, X_c] \right)\bigg|_{\text{D5}} = 0. \tag{3.15b}$$

### 3.1.2 NS5 Boundary

Similar to (3.9), at the NS5 boundary the $\vec{X}$ fields must satisfy Dirichlet boundary condition:

$$\vec{X}|_{\text{NS5}} = 0. \tag{3.16}$$

The constant value 0 corresponds to locations of NS5 branes in $\mathbb{R}^3_{789}$ directions. For now we assume the NS5 branes to be at the origin, but later we shall separate them in the $\mathbb{R}^3_{789}$ direction shifting the above boundary condition. (3.16) satisfies (3.8b) and (3.8d). The remaining equations are satisfied if we impose at the NS5 boundary:

$$\epsilon_{\lambda\mu\nu}F^{3\lambda} + \gamma'F_{\mu\nu} = 0, \tag{3.17a}$$

$$D_3 Y_a + \frac{u'}{2}\epsilon_{abc}[Y_b, Y_c] = 0, \tag{3.17b}$$

for some numbers $\gamma'$ and $u'$ along with:

$$\bar{\varepsilon}_0(1 + \gamma' B_0)\vartheta' = 0, \tag{3.18a}$$

$$\bar{\varepsilon}_0 B_2 \vartheta' = 0, \tag{3.18b}$$

$$\bar{\varepsilon}_0(1 + u' B_1)\vartheta' = 0. \tag{3.18c}$$

By assuming the same form for $\varepsilon_0$ as in (3.12) we find the following expressions for $\gamma'$, $u'$, and $\vartheta'$ from (3.18):

$$\gamma' = \frac{2a}{1 - a^2}, \qquad u' = \frac{2a}{1 + a^2}, \qquad \vartheta' = \begin{pmatrix} -a \\ 1 \end{pmatrix}. \tag{3.19}$$

In the D5 case, the choice $a = \infty$ leads to the standard Dirichlet equations (3.15) for the bosonic fields. This special value of $a$ leads to $\gamma', u' \to 0$ and thus the following boundary conditions for the gauge field and $\vec{Y}$ at the NS5 boundary preserve the same supersymmetry as (3.15):

$$F_{3\lambda}|_{\text{NS5}} = 0, \tag{3.20a}$$

$$D_3 Y_a|_{\text{NS5}} = 0. \tag{3.20b}$$

We recognize them as the standard Neumann boundary conditions for the gauge field and the scalar fields.

## 3.2 Topological Twists Preserved by $1/2$-BPS Boundary Conditions

In order to get 4d CS theory from the D5 branes of Table 1, we first need to perform a supersymmetric twist of the type IIB string theory background which induces the holomorphic-topological twist of the 6d $\mathcal{N} = (1,1)$ SYM describing the D5 brane dynamics [23, §6.1]. What is the effect of this twisted string theory background on the 4d $\mathcal{N} = 4$ theory describing the D3 branes? In [26], Kapustin and Witten defined a family of topological twists of 4d $\mathcal{N} = 4$ SYM parameterized by a complex number $t \in \mathbb{CP}^1$. The aforementioned twisted string theory induces precisely the Kapustin-Witten (KW) twist corresponding to $t = i$ [40, §6.2]. However, the analysis of [40] only looked at the bulk 4d theory and not at the boundary conditions. So, for consistency, we must check that this particular twist is preserved by the boundary conditions (3.9), (3.15), (3.16), and (3.20) given by the choice $a = \infty$. The question of KW twists preserved by $1/2$-BPS boundary conditions was addressed in [41]. We recall the setup to establish notation.

The topological twist is most naturally performed in Euclidean signature. We therefore Wick rotate $\mathbb{R}^{1,9}$ to $\mathbb{R}^{10}$. Essentially the only important change from our earlier discussion is the chirality constraint on an SO(10) spinor, which changes from (3.1) to:

$$\Gamma_{0\dots9}\varepsilon = -i\varepsilon\,, \tag{3.21}$$

where $\Gamma_I$ now refers to generators of the Clifford algebra associated to SO(10).

We start by viewing $\mathbb{R}^8_{0\dots7}$ as the cotangent bundle of $\mathbb{R}^4_{0123}$. The choice of the subspace $\mathbb{R}^4_{4567}$ of $\mathbb{R}^4_{4\dots9}$ as the cotangent directions explicitly breaks the SO(6) R-symmetry of 4d $\mathcal{N} = 4$ supersymmetry down to SO(4) × SO(2) where SO(4) and SO(2) acts on $\mathbb{R}^4_{4567}$ and $\mathbb{R}^2_{89}$ respectively. Let us denote the rotation groups of $\mathbb{R}^4_{0123}$ and $\mathbb{R}^4_{4567}$ by SO(4) and SO(4)$_R$ respectively. We then pick an isomorphism $\varkappa : SO(4) \xrightarrow{\sim} SO(4)_R$ and declare the image of the diagonal embedding $(1 \times \varkappa) : SO(4) \to SO(4) \times SO(4)_R$ to be the new space-time isometry. We denote this new isometry by SO(4)$'$. Up to an inconsequential relabeling of coordinates, we can concretely write down the action of $\varkappa$ on the generators of SO(4) in terms of the following even elements of the 10d Clifford algebra:

$$\varkappa_* : \mathfrak{so}(4) \xrightarrow{\sim} \mathfrak{so}(4)_R\,, \qquad \varkappa_* : \Gamma_{\mu\nu} \mapsto \Gamma_{\mu+4,\nu+4}\,, \qquad \mu, \nu \in \{0,1,2,3\}\,. \tag{3.22}$$

This redefinition of space-time isometry reduces the bosonic symmetry from SO(4) × SO(6) to

$$J := SO(4)' \times SO(2)\,. \tag{3.23}$$

The spin representation $\mathbf{4}$ of SO(6)$_R$ transforms as $(\mathbf{2},\mathbf{1})^1 \oplus (\mathbf{1},\mathbf{2})^{-1}$ under SO(4)$_R$ × SO(2).[8] Similarly, $\overline{\mathbf{4}}$ transforms as the complex conjugate $(\mathbf{2},\mathbf{1})^{-1} \oplus (\mathbf{1},\mathbf{2})^1$. Therefore, taking into account the identification of SO(4) isometry and SO(4)$_R$, the spin representations $S_\ell$ and $S_r$ from (3.4) transform under $J$ as:

$$S_\ell = (\mathbf{1},\mathbf{1})^{-1} \oplus (\mathbf{3},\mathbf{1})^{-1} \oplus (\mathbf{2},\mathbf{2})^1\,, \qquad S_r = (\mathbf{1},\mathbf{1})^{-1} \oplus (\mathbf{1},\mathbf{3})^{-1} \oplus (\mathbf{2},\mathbf{2})^1\,. \tag{3.24}$$

We see that there are exactly two supercharges that transform as scalars under the twisted isometry SO(4)$'$. One of these two supercharges was left handed in the untwisted theory and the other right handed. Let us denote the spinors parameterizing these supercharges by $\varepsilon_\ell$ and $\varepsilon_r$ respectively. Define an arbitrary complex linear combination of $\varepsilon_\ell$ and $\varepsilon_r$:

$$\widetilde{\varepsilon} := u\varepsilon_\ell + v\varepsilon_r\,. \tag{3.25}$$

The scalar supercharge parameterized by $\widetilde{\varepsilon}$ squares to zero modulo gauge transformation. Twisting now simply means passing to the cohomology of this scalar nilpotent supercharge.

---

[8]The superscript refers to the SO(2) charge.

The choice of $\widetilde{\varepsilon}$ is defined up to an irrelevant scaling. Therefore, the true parameter for the choice of a scalar supercharge is the ratio

$$t := \frac{v}{u} \in \mathbb{CP}^1. \tag{3.26}$$

Here we allow the value $\infty$ for the possibility that $\widetilde{\varepsilon} = \varepsilon_r$. This $t$ is the complex parameter for the family of KW twists.

The question we want answered is: does the choice $a = \infty$ (from (3.12)) preserve the KW twist $t = i$? Witten showed in [41] that $t$ and $a$ are nicely related:[9]

$$a = i\frac{t+i}{t-i}. \tag{3.28}$$

In particular, we find that the choice $a = \infty$, which gives us the standard Dirichlet and Neuman boundary conditions, preserves precisely the topological twist labeled by $t = i$:

$$a = \infty \quad \Longleftrightarrow \quad t = i. \tag{3.29}$$

This twist is of the B-type [26], i.e., we can view the twisted 4d theory as a 2d B-model on $\mathbb{R}^2_{12}$ with fields valued in some appropriate infinite dimensional space. This of course is the twist which, once $\Omega$-deformation is turned on, will reduce to 2d BF theory on $\mathbb{R}^2_{03}$ – as we show below.

# 4 $\Omega$-deformed Kapustin-Witten (KW) Theory and BF Phase Spaces

Now that we know precisely which twist to use, we can look at the specific twisted theory and perform $\Omega$-deformation [42–44]. We shall find that the KW theory in a B-type $\Omega$-background localizes to 2d BF theory. While this is certainly known in literature [40, 45, 46], some details have yet to appear, so for completion we go through the intermediate steps in this section. First we look at the bulk theory, disregarding the boundaries, and then we look at the boundary conditions in terms of the fields of the twisted theory.

## 4.1 KW Theory as a 2d B-model

In the untwisted 4d $\mathcal{N} = 4$ SYM, the six adjoint scalars $\phi_4, \cdots, \phi_9$ transform as the vector of SO(6). Under the twisted space-time symmetry SO(4)$'$ the four scalars $\phi_4, \cdots, \phi_7$ transform as a 1-form:

$$\phi := \phi_4 dx^0 + \phi_5 dx^1 + \phi_6 dx^2 + \phi_7 dx^3. \tag{4.1}$$

This 1-form can be combined with the connection to form a complexified connection:

$$\mathcal{A}_\mu := A_\mu + i\phi_{\mu+4}, \qquad \overline{\mathcal{A}}_\mu := A_\mu - i\phi_{\mu+4} \qquad \mu \in \{0, 1, 2, 3\}. \tag{4.2}$$

---

[9]This relation is a consequence of comparing the constraints on the supersymmetries preserved by the boundary conditions and the twisting procedure. The constraints that $\varepsilon_\ell$ and $\varepsilon_r$ are scalars under SO(4)$'$ can be written as $(1 + \varkappa_*)(\Gamma_{\mu\nu})(\varepsilon_\bullet) = (\Gamma_{\mu\nu} + \Gamma_{\mu+4,\nu+4})\varepsilon_\bullet = 0$ for $\bullet \in \{\ell, r\}$. Combining these with the 10d chirality constraint (3.21) and the 4d chirality constraints $\Gamma_{0123}\varepsilon_\ell = -\varepsilon_\ell$, $\Gamma_{0123}\varepsilon_r = \varepsilon_r$ we can get:

$$\left(1 + i\frac{1-t^2}{1+t^2}B_0 + \frac{2t}{1+t^2}B_1\right)(\varepsilon_\ell + t\varepsilon_r) = 0. \tag{3.27}$$

It can be shown that $\varepsilon_0$, as defined in (3.12), satisfies the same equation iff $a$ and $t$ are related by (3.28). The overall sign of $a$ is different in (3.28) relative to [41] because our definition of $a$ in (3.13) differs by a sign.

The remaining two scalars $\phi_8$ and $\phi_9$ transform under the vector representation of SO(2), we repackage them into a complex scalar and its conjugate with definite SO(2) charges:

$$\sigma := \phi_8 - \mathrm{i}\phi_9 \,, \qquad \overline{\sigma} := \phi_8 + \mathrm{i}\phi_9 \,. \qquad (4.3)$$

Let us denote by $Q_\ell$ and $Q_r$ the supercharges parameterized by $\varepsilon_\ell$ and $\varepsilon_r$. The supercharge parameterized by $u\varepsilon_\ell + v\varepsilon_r$ is then:

$$Q(t) = uQ_\ell + vQ_r \,. \qquad (4.4)$$

As mentioned earlier, the supersymmetry generated by $Q(t)$ depends only on the ratio $t = v/u$. We denote the supersymmetry variation of a field $\Phi$ generated by $Q(\mathrm{i})$ as:

$$\delta_T \Phi := [Q(\mathrm{i}), \Phi\} \,. \qquad (4.5)$$

The bracket is anti-symmetric/symmetric if $\Phi$ is bosonic/fermionic.

The vector multiplet in the KW theory contains the following fields:

| | $\Omega^0(\mathbb{R}^4, \mathfrak{g}_\mathbb{C})$ | $\Omega^1(\mathbb{R}^4, \mathfrak{g}_\mathbb{C})$ | $\Omega^2(\mathbb{R}^4, \mathfrak{g}_\mathbb{C})$ |
|---|---|---|---|
| Bosonic | $P, \sigma, \overline{\sigma}$ | $\mathcal{A}, \overline{\mathcal{A}}$ | |
| Fermionic | $\eta, \overline{\eta}$ | $\psi, \overline{\psi}$ | $\chi$ |

$$(4.6)$$

where $\mathbb{R}^4 = \mathbb{R}^4_{0123}$ is the space-time of the 4d theory. With the addition of the auxiliary field $P$ and for $t = \mathrm{i}$ the supersymmetry transformation $\delta_T$ is nilpotent off-shell:

$$\delta_T^2 = 0 \,. \qquad (4.7)$$

This auxiliary field can be integrated out by adding an irrelevant $\delta_T$-exact term to the action (see §3.4 of [26]). The fermionic fields all come from the fermions of the 4d $\mathcal{N} = 4$ vector multiplet. The untwisted vector multiplet contains fermions in the representation $S_\ell \oplus S_r$ which is the same as in (3.4). In the twisted theory this representation breaks down as representation of the twisted space-time symmetry SO(4)$'$ as in (3.24). We see that there are two scalars, a self-dual 2-form, an anti self-dual 2-form, and two 1-forms. The self-dual and anti self-dual forms combine to make up a complete 2-form. We see these fields in the above field content. Variations of the fields under the twisting supercharge are:

$$
\begin{aligned}
\delta_T \mathcal{A} &= 0 \,, & \delta_T \overline{\psi} &= \mathcal{D}\sigma \,, \\
\delta_T \overline{\mathcal{A}} &= \psi \,, & \delta_T \psi &= 0 \,, \\
\delta_T \sigma &= 0 \,, & \delta_T \eta &= P \,, \\
\delta_T \overline{\sigma} &= \overline{\eta} \,, & \delta_T \overline{\eta} &= 0 \\
\delta_T P &= 0 \,, & \delta_T \chi &= \mathcal{F} \,.
\end{aligned}
\qquad (4.8)
$$

Here the covariant derivative is defined with respect to the complex connection:

$$\mathcal{D} = \mathrm{d} + \mathcal{A} \wedge \,, \qquad (4.9)$$

and $\mathcal{F} = \mathrm{d}\mathcal{A} + \frac{1}{2}\mathcal{A} \wedge \mathcal{A}$ is the curvature of $\mathcal{A}$.[10] $\delta_T$ clearly satisfies (4.7).

---

[10]We need to define the wedge product involving Lie algebra valued forms as our convention differs from many standard literature. Let $\mathfrak{g}$ be any Lie algebra and $M$ a $\mathfrak{g}$-module with the Lie algebra homomorphism $\rho : \mathfrak{g} \to \mathrm{End}(M)$. Let $a = a_{I_1 \cdots I_p} \mathrm{d}x^{I_1} \wedge \cdots \wedge \mathrm{d}x^{I_p}$ be a $\mathfrak{g}$-valued $p$-form and $v = v_{J_1 \cdots J_q} \mathrm{d}x^{J_1} \wedge \cdots \wedge \mathrm{d}x^{J_q}$ an $M$-valued $q$-form. Then we define the wedge product between them as:

$$a \wedge v := \rho(a_{I_1 \cdots I_p})(v_{J_1 \cdots J_q}) \mathrm{d}x^{I_1} \wedge \cdots \wedge \mathrm{d}x^{I_p} \wedge \mathrm{d}x^{J_1} \wedge \cdots \wedge \mathrm{d}x^{J_q} \,. \qquad (4.10)$$

In particular, for a Lie algebra valued connection $\mathcal{A}_\mu \mathrm{d}x^\mu$ – thinking of $\mathfrak{g}$ as the adjoint module of $\mathfrak{g}$ – we get from the above definition $\mathcal{A} \wedge \mathcal{A} = [\mathcal{A}_\mu, \mathcal{A}_\nu] \mathrm{d}x^\mu \wedge \mathrm{d}x^\nu$. This differs from many literature where $\mathcal{A} \wedge \mathcal{A}$ only involves the associative product in the universal enveloping algebra of the Lie algebra which then needs to be anti-symmetrized to get the Lie bracket. This is the reason for the factor of half in our expression for the curvature which may seem unfamiliar to some.

We shall write down the bosonic part of the action of the KW theory (in (4.44)) after we introduce a 2d formulation of the theory.

### 4.1.1   2d Multiplets and Supersymmetry

The 4d KW theory on $\mathbb{R}^4_{0123}$ can be viewed as a 2d B-model on $\mathbb{R}^2_{12}$ with an infinite dimensional gauge group and fields valued in certain infinite dimensional target spaces. More specifically, the 2d vector multiplet contains the connections

$$\mathcal{A}^{(2)} := \mathcal{A}_i \mathrm{d}x^i, \qquad \overline{\mathcal{A}}^{(2)} := \overline{\mathcal{A}}_i \mathrm{d}x^i. \tag{4.11}$$

In this section the indices $i, j, \cdots$ run over 1 and 2. Viewed as 1-forms on $\mathbb{R}^2_{12}$, they are valued in the infinite dimensional Lie algebra:

$$\mathfrak{G}_{\mathbb{C}} := \Omega^0(\mathbb{R}^2_{03}, \mathfrak{g}_{\mathbb{C}}). \tag{4.12}$$

The Lie bracket of $\mathfrak{G}_{\mathbb{C}}$ consists of point-wise multiplication on $\mathbb{R}^2_{03}$ and the ordinary Lie bracket on $\mathfrak{g}_{\mathbb{C}}$. The full 2d B-model vector multiplet contains the following fields:

|  | $\Omega^0(\mathbb{R}^2_{12}, \mathfrak{G}_{\mathbb{C}})$ | $\Omega^1(\mathbb{R}^2_{12}, \mathfrak{G}_{\mathbb{C}})$ | $\Omega^2(\mathbb{R}^2_{12}, \mathfrak{G}_{\mathbb{C}})$ |
|---|---|---|---|
| Bosonic | $P$ | $\mathcal{A}^{(2)}, \overline{\mathcal{A}}^{(2)}$ | |
| Fermionic | $\eta$ | $\psi^{(2)} := \psi_i \mathrm{d}x^i$ | $\chi^{(2)} := \frac{1}{2}\chi_{ij}\mathrm{d}x^i \wedge \mathrm{d}x^j$ |

$$\tag{4.13}$$

B-model supersymmetry transformations of these fields [47] coincide with the KW supersymmetry (4.8) restricted to these fields:

$$\begin{aligned}
\delta_T \mathcal{A}^{(2)} &= 0, & \delta_T \overline{\mathcal{A}}^{(2)} &= \psi^{(2)}, \\
\delta_T \psi^{(2)} &= 0, & \delta_T \eta &= P, \\
\delta_T P &= 0, & \delta_T \chi^{(2)} &= \mathcal{F}^{(2)}.
\end{aligned} \tag{4.14}$$

Here $\mathcal{F}^{(2)} = \mathrm{d}^{(2)}\mathcal{A}^{(2)} + \frac{1}{2}\mathcal{A}^{(2)} \wedge \mathcal{A}^{(2)}$ is the curvature of $\mathcal{A}^{(2)}$.

The remaining two components of the 4d connections, namely:

$$\mathcal{A}^{(c)} := \mathcal{A}^{(c)}_m \mathrm{d}x^m, \qquad \overline{\mathcal{A}}^{(c)} := \overline{\mathcal{A}}^{(c)}_m \mathrm{d}x^m, \tag{4.15}$$

belong to 2d chiral multiplets. Here $m, n, \cdots$ run over 0 and 3. The fields $\mathcal{A}^{(c)}$ and $\overline{\mathcal{A}}^{(c)}$ are scalars on $\mathbb{R}^2_{12}$ – the world-volume of the B-model – and they are $\mathfrak{g}_{\mathbb{C}}$ valued 1-forms on $\mathbb{R}^2_{03}$. So the chiral multiplet is valued in the infinite dimensional space:

$$\mathfrak{X} := \Omega^1(\mathbb{R}^2_{03}, \mathfrak{g}_{\mathbb{C}}). \tag{4.16}$$

The full content of this chiral multiplet is as follows:

|  | $\Omega^0(\mathbb{R}^2_{12}, \mathfrak{X})$ | $\Omega^1(\mathbb{R}^2_{12}, \mathfrak{X})$ | $\Omega^2(\mathbb{R}^2_{12}, \mathfrak{X})$ |
|---|---|---|---|
| Bosonic | $\mathcal{A}^{(c)}, \overline{\mathcal{A}}^{(c)}$ | | $\mathsf{F}, \overline{\mathsf{F}}$ |
| Fermionic | $\psi^{(c)} := \psi_m \mathrm{d}x^m$ | $\chi^{(c)} := \chi_{im}\mathrm{d}x^i \wedge \mathrm{d}x^m$ | $\overline{\mathsf{M}}$ |

$$\tag{4.17}$$

We have introduced the auxiliary fields $\mathsf{F}, \overline{\mathsf{F}}, \overline{\mathsf{M}} \in \Omega^2(\mathbb{R}^2_{12}, \mathfrak{X}) \subset \Omega^3(\mathbb{R}^4_{0123}, \mathfrak{g}_{\mathbb{C}})$. To compare the B-model with the KW theory we will need to integrate out the bosonic auxiliary fields $\mathsf{F}$ and $\overline{\mathsf{F}}$. After integrating out these two fields we will find that

$$\overline{\mathsf{M}}_{12} = -\psi_m \mathrm{d}x^m. \tag{4.18}$$

B-model supersymmetry transformations of these chiral multiplet fields are:

$$
\begin{aligned}
\delta_T \mathcal{A}^{(c)} &= 0, & \delta_T \overline{\mathcal{A}}^{(c)} &= \psi^{(c)}, \\
\delta_T \chi^{(c)} &= \mathcal{F}_{im} \mathrm{d}x^i \wedge \mathrm{d}x^m, & \delta_T \psi^{(c)} &= 0, \\
\delta_T \mathsf{F} &= \mathcal{D}^{(2)} \chi^{(c)} + \mathcal{D}^{(c)} \chi^{(2)} & \delta_T \overline{\mathsf{M}} &= \overline{\mathsf{F}}, \\
& & \delta_T \overline{\mathsf{F}} &= 0.
\end{aligned}
\tag{4.19}
$$

The differentials are defined as:

$$
\mathcal{D}^{(2)} := \mathrm{d}x^i \partial_{x^i} + \mathcal{A}^{(2)} \wedge, \qquad \mathcal{D}^{(c)} := \mathrm{d}x^m \partial_{x^m} + \mathcal{A}^{(c)} \wedge.
\tag{4.20}
$$

These transformations coincide with the restriction of the KW supersymmetry to the above fields, excluding the newly introduced auxiliary fields.

The remaining bosonic fields of the KW theory, namely $\sigma$ and $\overline{\sigma}$ fit into another $\mathfrak{G}_{\mathbb{C}}$ valued chiral multiplet with the following field content:

|  | $\Omega^0(\mathbb{R}^2_{12}, \mathfrak{G}_{\mathbb{C}})$ | $\Omega^1(\mathbb{R}^2_{12}, \mathfrak{G}_{\mathbb{C}})$ | $\Omega^2(\mathbb{R}^2_{12}, \mathfrak{G}_{\mathbb{C}})$ |
|---|---|---|---|
| Bosonic | $\sigma, \overline{\sigma}$ |  | $\mathsf{G}, \overline{\mathsf{G}}$ |
| Fermionic | $\overline{\eta}$ | $\overline{\psi}^{(c)} := \overline{\psi}_m \mathrm{d}x^m$ | $\overline{\mathsf{N}}$ |

(4.21)

Similar to the previous chiral multiplet, we have introduced auxiliary fields $\mathsf{G}, \overline{\mathsf{G}}$ and $\overline{\mathsf{N}}$. We integrate out $\mathsf{G}$ and $\overline{\mathsf{G}}$, after which we are lead to the identification

$$
\overline{\mathsf{N}}_{12} = \chi_{03} \mathrm{d}x^0 \wedge \mathrm{d}x^3.
\tag{4.22}
$$

The supersymmetry transformations of these fields are:

$$
\begin{aligned}
\delta_T \sigma &= 0, & \delta_T \overline{\sigma} &= \overline{\eta}, \\
\delta_T \overline{\psi}^{(c)} &= \mathcal{D}^{(2)} \sigma, & \delta_T \overline{\eta} &= 0, \\
\delta_T \mathsf{G} &= \mathcal{D}^{(2)} \overline{\psi}^{(c)} + \sigma \wedge \chi^{(2)}, & \delta_T \overline{\mathsf{N}} &= \overline{\mathsf{G}}, \\
& & \delta_T \overline{\mathsf{G}} &= 0.
\end{aligned}
\tag{4.23}
$$

The transformations (4.14), (4.19), and (4.23) satisfy the topological supersymmetry algebra:

$$
\delta_T^2 = 0,
\tag{4.24}
$$

the same as (4.7).

Taking into account the on-shell identifications (4.18) and (4.22), which we will justify in (4.39) and (4.43) respectively, all the fields of the 4d KW theory have been encoded into various 2d B-model multiplets. Let us now look at the actions for these multiplets.

### 4.1.2 Actions

We define the following inner product on the Lie algebra $\mathfrak{G}_{\mathbb{C}}$: for $a, b \in \mathfrak{G}_{\mathbb{C}} = \Omega^0(\mathbb{R}^2_{03}, \mathfrak{g}_{\mathbb{C}})$

$$
\mathrm{tr}(ab) = \int_{\mathbb{R}^2_{03}} \star_{03} \mathrm{tr}(ab) = \int_{\mathbb{R}^2_{03}} \mathrm{d}x^0 \wedge \mathrm{d}x^3 \, \mathrm{tr}(ab),
\tag{4.25}
$$

here $\star_{03}$ is the Hodge star operator on $\mathbb{R}^2_{03}$. The 2d vector multiplet action can be written as:

$$
S_V = \int_{\mathbb{R}^2_{12}} \delta_T \mathrm{tr}\left( \left(- \star_{12} P + 2\mathrm{i}D^{(2)} \star_{12} \phi^{(2)}\right)\eta - \chi^{(2)} \star_{12} \overline{\mathcal{F}}^{(2)} \right)
$$

$$= \int_{\mathbb{R}^4_{0123}} \mathrm{d}^4x \, \mathrm{tr}\left( (-P + 2\mathrm{i}D^i\phi_i)P - \frac{1}{2}\mathcal{F}_{ij}\overline{\mathcal{F}}^{ij} \right) + \cdots \tag{4.26}$$

The ellipses hide terms involving fermions. We shall often ignore such terms for the sake of simplicity. Note that the above action, when viewed as a 4d action, contains kinetic terms for only $\mathcal{A}_1$ and $\mathcal{A}_2$. It is missing the kinetic terms for $\mathcal{A}_0$ and $\mathcal{A}_3$. These components are part of a 2d chiral multiplet (4.17) and therefore their kinetic term will come from 2d chiral multiplet actions, as we shall now see.

In addition to the inner product (4.25) on $\mathfrak{G}_{\mathbb{C}}$, we also need an inner product on $\mathfrak{X}$ (4.16) to write down an action for the chiral multiplets. For $a, b \in \Omega^p(\mathbb{R}^2_{03}, \mathfrak{g}_{\mathbb{C}})$ define the inner product:

$$\langle a, b \rangle := \int_{\mathbb{R}^2_{03}} \mathrm{d}^2x \, \mathrm{tr}(a \wedge \star_{03} b). \tag{4.27}$$

We can then write down the action for the first chiral multiplet (4.17) as follows:

$$S_{C_1} = \int_{\mathbb{R}^2_{12}} \delta_T \left( \left\langle -\overline{\mathcal{F}}_{im}\mathrm{d}x^i \wedge \mathrm{d}x^m, \star_{12}\chi_{im}\mathrm{d}x^i \wedge \mathrm{d}x^m \right\rangle + \left\langle 2\mathrm{i}D^{(c)} \star_{03} \phi^{(c)}, \star_{12}\eta \right\rangle - \left\langle \overline{\mathsf{M}}, \star_{12}\mathsf{F} \right\rangle \right)$$

$$= \int_{\mathbb{R}^4_{0123}} \mathrm{d}^4x \, \mathrm{tr}\left( -\overline{\mathcal{F}}^{im}\mathcal{F}_{im} + 2\mathrm{i}D_m\phi^m P - \overline{\mathsf{F}}_{m12}\mathsf{F}^{m12} \right) + \cdots \tag{4.28}$$

and that of the second chiral multiplet (4.21) as:

$$S_{C_2} = \int_{\mathbb{R}^2_{12}} \delta_T \left( -\left\langle \overline{\mathcal{D}}^{(2)}\overline{\sigma}, \star_{12}\overline{\psi}^{(c)} \right\rangle + \left\langle [\sigma, \overline{\sigma}], \star_{12}\eta \right\rangle - \left\langle \overline{\mathsf{N}}, \star_{12}\mathsf{G} \right\rangle \right)$$

$$= \int_{\mathbb{R}^4_{0123}} \mathrm{d}^4x \, \mathrm{tr}\left( -\overline{\mathcal{D}}^i\overline{\sigma}\mathcal{D}_i\sigma + [\sigma, \overline{\sigma}]P - \overline{\mathsf{G}}_{12}\mathsf{G}^{12} \right) + \cdots \tag{4.29}$$

As before, we have omitted terms containing fermions in the final expressions.

To complete the 2d theory we also need a superpotential. We shall find that we recover the KW theory precisely with the following superpotential:

$$W(\sigma, \mathcal{A}^{(c)}) := \left\langle \sigma, \star_{03}\mathcal{F}^{(c)} \right\rangle = \int_{\mathbb{R}^2_{03}} \mathrm{tr}\left( \sigma \left( \mathrm{d}^{(c)}\mathcal{A}^{(c)} + \frac{1}{2}\mathcal{A}^{(c)} \wedge \mathcal{A}^{(c)} \right) \right). \tag{4.30}$$

This leads to the superpotential action:

$$S_W = \int_{\mathbb{R}^2_{12}} \left\langle \mathsf{F}, \frac{\delta W}{\delta \mathcal{A}^{(c)}} \right\rangle - \left\langle \overline{\mathsf{F}}, \frac{\delta \overline{W}}{\delta \overline{\mathcal{A}}^{(c)}} \right\rangle + \left\langle \mathsf{G}, \frac{\delta W}{\delta \sigma} \right\rangle - \left\langle \overline{\mathsf{G}}, \frac{\delta \overline{W}}{\delta \overline{\sigma}} \right\rangle + \cdots$$

$$= \int_{\mathbb{R}^4_{0123}} \mathrm{d}^4x \, \mathrm{tr}(-\mathsf{F}^m{}_{12}\mathcal{D}_m\sigma + \overline{\mathsf{F}}^m{}_{12}\overline{\mathcal{D}}_m\overline{\sigma} + \mathsf{G}_{12}\mathcal{F}^{03} - \overline{\mathsf{G}}_{12}\overline{\mathcal{F}}^{03}) + \cdots \tag{4.31}$$

The total B-model action is the sum of (4.26), (4.28), (4.29), and (4.31):

$$S_{\mathrm{B}}^{\mathrm{2d}} = S_V + S_{C_1} + S_{C_2} + S_W. \tag{4.32}$$

The matching of the 2d B-model action and the 4d KW action will occur on shell. We need to integrate out the bosonic auxiliary fields. Terms in (4.32) containing $P$ are:

$$\int_{\mathbb{R}^4_{0123}} \mathrm{d}^4x \, \mathrm{tr}\left( -P^2 + 2\left( \mathrm{i}D^i\phi_i + \mathrm{i}D_m\phi^m + \frac{1}{2}[\sigma, \overline{\sigma}] \right)P \right). \tag{4.33}$$

Therefore, $P$ can be integrated out by the following equation of motion:

$$P = iD^\mu \phi_\mu + \frac{1}{2}[\sigma, \overline{\sigma}]. \tag{4.34}$$

The effect of integrating $P$ out is to replace the terms (4.33) in the action involving $P$ with the following term:

$$\int_{\mathbb{R}^4_{0123}} d^4x \, \mathrm{tr}\left( iD^\mu \phi_\mu + \frac{1}{2}[\sigma, \overline{\sigma}] \right)^2. \tag{4.35}$$

Terms in the action involving $\mathsf{F}, \overline{\mathsf{F}}$ are:

$$\int_{\mathbb{R}^4_{0123}} d^4x \, \mathrm{tr}\left( -\overline{\mathsf{F}}_{m12}\mathsf{F}^{m12} - \mathsf{F}^m{}_{12}\mathcal{D}_m\sigma + \overline{\mathsf{F}}^m{}_{12}\overline{\mathcal{D}}_m\overline{\sigma} \right). \tag{4.36}$$

Their equations of motion are:

$$\overline{\mathsf{F}}_{m12} = -\mathcal{D}_m\sigma, \qquad \mathsf{F}_{m12} = \overline{\mathcal{D}}_m\overline{\sigma}. \tag{4.37}$$

By integrating out $\mathsf{F}, \overline{\mathsf{F}}$ we replace the action (4.36) by:

$$-\int_{\mathbb{R}^4_{0123}} d^4x \, \mathrm{tr}\left( \overline{\mathcal{D}}^m\overline{\sigma}\mathcal{D}_m\sigma \right). \tag{4.38}$$

Also note that, after integrating out these fields the supersymmetry variation of $\overline{\mathsf{M}}$ (4.19) becomes:

$$\delta_T\overline{\mathsf{M}}_{12} = \overline{\mathsf{F}}_{12} = -\mathcal{D}_m\sigma. \tag{4.39}$$

Comparing this with the supersymmetry transformation of $\overline{\psi}$ in the KW theory (4.8) leads to the identification (4.18).

Terms in the action involving $\mathsf{G}, \overline{\mathsf{G}}$ are:

$$\int_{\mathbb{R}^4_{0123}} d^4x \, \mathrm{tr}\left( -\overline{\mathsf{G}}_{12}\mathsf{G}^{12} + \mathsf{G}_{12}\mathcal{F}^{03} - \overline{\mathsf{G}}_{12}\overline{\mathcal{F}}^{03} \right). \tag{4.40}$$

Their equations of motion:

$$\overline{\mathsf{G}}_{12} = \mathcal{F}_{03}, \qquad \mathsf{G}^{12} = -\overline{\mathcal{F}}^{03}. \tag{4.41}$$

By integrating out $\mathsf{G}, \overline{\mathsf{G}}$ we replace the action (4.40) by:

$$-\int_{\mathbb{R}^4_{0123}} d^4x \, \mathcal{F}_{03}\overline{\mathcal{F}}^{03}. \tag{4.42}$$

Integrating out $\overline{\mathsf{G}}$ makes the on shell supersymmetry of $\overline{\mathsf{N}}$ (4.23):

$$\delta_T\overline{\mathsf{N}}_{12} = \overline{\mathsf{G}}_{12} = \mathcal{F}_{03}. \tag{4.43}$$

Comparing this with the supersymmetry of the KW theory (4.8) we find the identification (4.22).

Thus, after integrating out all the auxiliary fields, the B-model action becomes:

$$S_{\mathrm{B}}^{2d} \xrightarrow{\text{on shell}} \int_{\mathbb{R}^4_{0123}} d^4x \, \mathrm{tr}\left( -\frac{1}{2}\mathcal{F}_{\mu\nu}\overline{\mathcal{F}}^{\mu\nu} - \overline{\mathcal{D}}^\mu\overline{\sigma}\mathcal{D}_\mu\sigma + \left( iD^\mu\phi_\mu + \frac{1}{2}[\sigma,\overline{\sigma}] \right)^2 + \cdots \right). \tag{4.44}$$

This is the fermion free part of the 4d KW action [26], or equivalently, the dimensional reduction of the holomorphic-topological twist of 6d $\mathcal{N} = (1,1)$ SYM – reduced along the holomorphic directions [23].

### 4.1.3  $\Omega$-deforming the B-model

A B-model on $\mathbb{R}^2$ with a U(1) symmetry generated by rotation of the space-time admits an $\Omega$-deformation [48]. Let $V = \hbar\partial_\theta$ be the generator of rotation where $\theta$ is the angular coordinate on $\mathbb{R}^2$ and $\hbar$ is a deformation parameter. The B-model supersymmetry variation $\delta_T$ can be deformed in a way such that instead of squaring to zero, it squares to the action of $V$:

$$\delta_T \rightsquigarrow \delta_V \quad \text{such that on fields} \quad \delta_V^2 = \mathcal{L}_V \text{ (modulo gauge transformations)}, \qquad (4.45)$$

where $\mathcal{L}_V$ is the Lie derivative with respect to the vector field $V$. The precise deformations of $\delta_T$ acting on the vector multiplet (4.14) and the chiral multiplets, (4.19) and (4.23), are given in (A.1), (A.2), and (A.3) respectively. Suppose that the B-model has chiral multiplets valued in some complex symplectic target $\Upsilon$, a compact gauge group $G$, and a $G_{\mathbb{C}}$-invariant holomorphic superpotential $W : \Upsilon \to \mathbb{C}$. Then the $\Omega$-deformed theory localizes to a matrix model with the action $\frac{1}{\hbar}W$ and the complexified gauge group $G_{\mathbb{C}}$. Path integrals in this matrix model are performed over a Lagrangian subspace of $\Upsilon$ – the choice of the Lagrangian depends on the choice of boundary conditions for the B-model fields at infinity of the space-time $\mathbb{R}^2$. Applying this process to the B-model description of the KW theory with compact gauge group $G$ and superpotential $W$ (4.30) from the last section we find that once $\Omega$-deformation is turned on with respect to the rotation of the $\mathbb{R}^2_{12}$ plane, the theory localizes to a 2d BF theory on $\mathbb{R}^2_{03}$ with gauge group $G_{\mathbb{C}}$ and the following action [28, 29]:

$$S_{\text{BF}} = \frac{1}{\hbar}W = \frac{1}{\hbar}\int_{\mathbb{R}^2_{03}} dx^0 \wedge dx^3 \operatorname{tr} \sigma \mathcal{F}_{03} . \qquad (4.46)$$

Varying this action we get the following equations of motion:

$$\mathcal{F}_{03} = 0 \quad \text{and} \quad \mathcal{D}^{(c)}\sigma = 0 , \qquad (4.47)$$

where the covariant derivative $\mathcal{D}^{(c)}$ only has differentials and connections in the directions of $\mathbb{R}^2_{03}$, as defined in (4.20).

## 4.2  Boundary Conditions in BF Theory

Instead of considering the BF theory on the infinite plane as in (4.46) we can put it on a strip $\mathbb{R} \times \mathbb{I}$ where we consider the $x^3$ direction to be a finite interval $\mathbb{I}$. We can choose a gauge where $\mathcal{A}_0 = 0$. In this gauge the second equation of motion from (4.47) says that $\sigma$ and $\mathcal{A}_3$ are independent of $x^0$ and they further satisfy:

$$\partial_3\sigma + [\mathcal{A}_3, \sigma] = 0 , \qquad (4.48)$$

which is $G_{\mathbb{C}}$-invariant. Appropriate boundary conditions must be imposed on $\sigma$ and gauge transformations at the boundary which we shall discuss in this section. For now we note that the moduli space of solutions to (4.48) modulo complex gauge transformations is the same, as a complex symplectic manifold, as the moduli space of solutions to Nahm's equation for the triple $(\operatorname{Im}\mathcal{A}_3, \operatorname{Re}\sigma, -\operatorname{Im}\sigma) = (\phi_7, \phi_8, \phi_9)$ modulo real gauge transformations [49].

Fields of the BF theory are the complex connection $\mathcal{A}_0 dx^0 + \mathcal{A}_3 dx^3$ and the complex adjoint scalar $\sigma$. These fields are written in terms of the fields of 4d $\mathcal{N} = 4$ SYM in (4.2) and (4.3). In the presence of a boundary, it is better to use the notation for the scalar fields suited to the symmetry of the boundary (3.3). We have:

$$\mathcal{A}_0 = A_0 + iY_1 , \qquad \mathcal{A}_3 = A_3 + iX_1 , \qquad \sigma = X_2 - iX_3 . \qquad (4.49)$$

We can now translate the 1/2-BPS boundary conditions of the 4d theory into boundary conditions of the BF theory. We can readily use the analysis of the moduli space of solutions to Nahm's equation as a complex space from [24]. In this reference the choice of complex structure was arbitrary and breaks the hyper-Kähler symmetry of the moduli space. This extended symmetry is broken in our case in the process of twisting when we broke the SO(6) R-symmetry of 4d $\mathcal{N} = 4$ SYM down to SO(4) × SO(2). This SO(2) acts on the moduli space of solutions to Nahm's equation and making the action holomorphic is equivalent to choosing a complex structure. The choice is made by declaring $\sigma$ to be a holomorphic coordinate.

We create boundaries for the BF theory by starting with 4d $\mathcal{N} = 4$ SYM on an interval and then turning on Ω-deformation. In terms of branes, We start with a configuration where D3 branes are suspended between five-branes. We consider two types of boundaries, one created by D5 branes, corresponding to the boundary conditions of §3.1.1 and the other created by NS5 branes corresponding to the boundary conditions of §3.1.2. Similar setups also appeared in [50] where the author shows using supersymmetric localization that interfaces in 4d $\mathcal{N} = 4$ SYM created by five-branes can be reduced to a 2d/1d setup involving interfaces in 2d Yang-Mills (YM) theory. This setup was further used in [51] to compute 1-point correlation functions in 2d YM in the presence of a D5-type interface. 2d BF theory is a zero coupling limit of 2d YM and these results should be relevant for defining traces of algebras coupled to the line operators of 4d CS that these defects create in our setup.

### 4.2.1 D5 Boundary

The most general D5 type boundary we consider comes from the brane configuration in Fig. 4. There are $n$ ordered D5 branes such that the $i$th D5 brane has linking number $K_i$. Since there

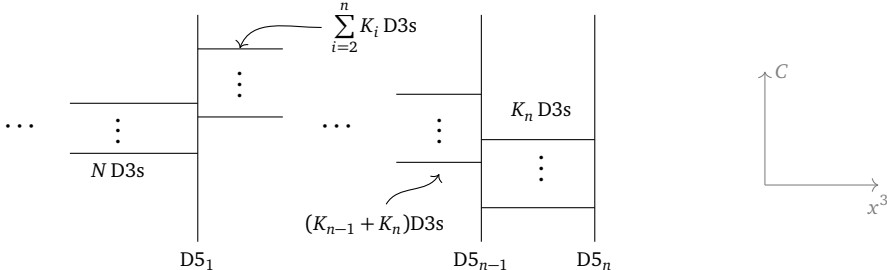

Figure 4: A generic D5 type boundary determined by $\mathbf{K} = (K_1, \cdots, K_n)$ and $K_i$ is the linking number of the $i$th D5 brane. We denote the total number of D3 branes by $N = \sum_{i=1}^n K_i$.

are no NS5 branes to the right of any D5 brane, $K_i$ coincides with the number of D3 branes ending on the $i$th D5 brane from the left according to the definition of linking numbers (2.2). In some other brane configuration related by Hanany-Witten transitions the number of D3 branes ending on the five-branes will generally be different but linking numbers are invariant under such transformations. The brane configuration of Fig. 4 can be tabulated as in Table 2. We have specified the positions of the branes in some of the directions. The $i$th D5 brane is located at $x^3 = t_i$, $x^4 = m_i$, $x^5 = x^6 = 0$. We need these branes to be located at the center of the $\mathbb{R}^2_{56}$ plane in order to preserve the U(1) symmetry that acts by rotating the $\mathbb{R}^2_{12}$ and the $\mathbb{R}^2_{56}$ planes and which we use to turn on the Ω-deformation. Our brane configuration must be invariant under this rotation. All D3 branes attached to the $i$th D5 brane must be located at $(m_i, 0, 0)$ in $\mathbb{R}^3_{456}$. In order to have a gauge theory on the D5 branes with the gauge group U($n$) we must ultimately take the limit where for all $i$ we have $t_i \to t$ and $m_i \to m$ for some real numbers $t$ and $m$. Having a nonzero $m$ would modify the D5 boundary condition on $\vec{Y}$

Table 2: Directions wrapped by the D3 and D5 branes from Fig. 4. We denote the D3 branes attached to the $i$th D5 brane by $\mathrm{D3}_i^\alpha$ for different values of $\alpha$. We have also noted that fluctuations of the D3 branes in $\mathbb{R}^3_{456}$ and $\mathbb{R}^3_{789}$ direction corresponds to the triples of adjoint scalars $\vec{Y}$ and $\vec{X}$ respectively in the 4d $\mathcal{N}=4$ $\mathrm{U}(N)$ SYM theory on the D3 branes. Here $N$ is the total number of D3 branes.

| | $\mathbb{R}^2_{+\hbar}$ | | | | $\mathbb{R}^2_{-\hbar}$ | | | $C$ | |
| | 0 | 1 | 2 | 3 | 4 | 5 | 6 | 7 | 8 | 9 |
|---|---|---|---|---|---|---|---|---|---|---|
| Scalar Fields | | | | | $Y_1$ | $Y_2$ | $Y_3$ | $X_1$ | $X_2$ | $X_3$ |
| $\mathrm{D5}_i$ | × | × | × | $t_i$ | $m_i$ | 0 | 0 | × | × | × |
| $\mathrm{D3}_i^\alpha$ | × | × | × | × | $m_i$ | 0 | 0 | | | |

(3.9) by putting the triple $(m,0,0) \otimes \mathbb{1}_N \in \mathbb{R}^3 \otimes Z(\mathfrak{gl}_N)$ of central elements on the right hand side instead of zero. We can set $m = t = 0$ by choosing the origin of our coordinate system appropriately. We have left the locations of the D3 branes in the $\mathbb{R}^3_{789}$ directions unspecified because they are not fixed by this brane configuration and are not part of the D5 type boundary conditions.

**Remark 4.1** (Further Comments on Twisted Masses). We will ultimately let the D3 branes in Fig. 4 end on NS5 branes located to the left of all the D5 branes. Then, with finite extension in the $x^3$ direction, the low energy dynamics of the D3 branes will be captured by a 3d $\mathcal{N}=4$ theory, the one we denoted as $T^\vee[\mathrm{U}(N)]_K^L$ in §2. In terms of this 3d theory, the $m_i$'s are real twisted masses. As we discussed in §2, we expect to find the Higgs branch of this 3d theory as the phase space for the line operators under consideration. Nonzero twisted masses would lift parts of the Higgs branch. Note furthermore that in 3d $\mathcal{N}=4$ mass parameters usually come in real triplets, in terms of our brane construction such a triplet of twisted masses would correspond to the locations of the D3 branes in the $\mathbb{R}^3_{456}$ directions. Since we want to preserve the rotational symmetry of the $\Omega$-background, we could only turn on one component of these masses.

The world-volume theory of the D3 branes is 4d $\mathcal{N}=4$ SYM with $\mathrm{U}(N)$ gauge symmetry. After twisting and turning on $\Omega$-deformation this becomes 2d BF theory with gauge group $\mathrm{GL}_N$ where

$$N = \sum_{i=1}^n K_i \,. \tag{4.50}$$

The 1/2-BPS boundary conditions for the 4d SYM fields are the D5 type conditions mentioned in (3.9), (3.15). The real scalars $\vec{X}$ satisfy Nahm's equation (3.15b). We can combine two of the three equations to write down the following equation for the complexified BF fields $\sigma$ and $\mathcal{A}_3$:

$$\mathcal{D}_3\sigma := \partial_3\sigma + [\mathcal{A}_3, \sigma] = 0 \,. \tag{4.51}$$

And instead of imposing the remaining real equation and then considering solutions to the equations of motion with boundary conditions modulo real gauge transformations, in the BF theory we only impose the complex equation and quotient by complex gauge transformations.

In the special cases when the integers $K_i$ satisfy the following inequality:

$$K_i \geq K_{i+1} \geq 0 \quad \text{for all} \quad 1 \leq i < n \,, \tag{4.52}$$

the boundary of Fig. 4 corresponds to a simple polar boundary condition on the real adjoint scalars $X_i$ of the 4d theory. Assuming that the boundary is located at $x^3 = 0$, the fields $X_i$

behave near the boundary as (see §3.5 of [24]):

$$\vec{X}(x^3) = \frac{\vec{t}}{x^3} + O(1).$$

(4.53)

where $\vec{t} = (t_1, t_2, t_3)$ are images of a standard $\mathfrak{su}(2)$ basis under a homomorphism $\rho_K : \mathfrak{su}(2) \to \mathfrak{u}(N)$ satisfying $[t_i, t_j] = i\epsilon_{ijk}t_k$. As the notation suggests, the homomorphism depends on the linking numbers $K$. The form (4.53) of the 4d fields becomes the following boundary behavior of the BF field $\sigma$:

$$\sigma(x^3) = \frac{t_-}{x^3} + O(1),$$

(4.54)

where $t_- := t_2 - it_3$ is a nilpotent element of $\mathfrak{gl}_N$. The homomorphism $\rho_K$ is such that $t_-$ has the following Jordan normal form:

$$\begin{pmatrix} v_{K_1} & & \\ & \ddots & \\ & & v_{K_n} \end{pmatrix},$$

(4.55)

where $v_K$ is a regular nilpotent matrix of size $K \times K$ in the normal form.

$$v_K = \begin{pmatrix} 0 & 1 & & & \\ & 0 & 1 & & \\ & & \ddots & \ddots & \\ & & & 0 & 1 \\ & & & & 0 \end{pmatrix}.$$

(4.56)

For arbitrary linking numbers that do not satisfy the inequality (4.52), such a simple boundary condition does not exist. Instead, we keep the D5 branes separated along the $x^3$-direction and treat them as interfaces between theories with possibly different gauge groups. The boundary conditions on the fields of the 4d theory at these interfaces were described in [24] and we can easily translate them into boundary conditions for the BF fields.

From Fig. 4, between the $D5_{i-1}$ and the $D5_i$ branes there are

$$N_i := \sum_{j=i}^{n} K_j,$$

(4.57)

D3 branes which lead to a 4d theory with $U(N_i)$ gauge group and consequently to a 2d BF theory with $GL_{N_i}$ gauge group on an interval. To the left of the $D5_1$ brane there are $N_1 = N$ D3 branes leading to the $GL_N$ BF theory. Thus the $D5_i$ brane creates an interface between two theories such that the rank of the gauge group jumps from $N_i$ to $N_{i+1}$ from left to right. The last, i.e. the $n$th D5 brane is just a special case of this such that it interfaces between a $GL_{N_n}$ BF theory, and a trivial/empty theory.

To make contact with the diagrammatic notations for bow varieties from [6] and [7] we draw the stack of D3 brane segments between two adjacent five-branes (of any kind) as a wavy line and label the line by the number of D3 branes in the stack, such as . We mark each D5 brane by a ✕. So we represent the boundary from Fig. 4 diagrammatically as in Fig. 5.

To discuss the boundary conditions at an interface, let us focus on the $i$th D5 brane between two BF theories with gauge groups $GL_{N_i}$ and $GL_{N_{i+1}}$, i.e., we are focusing on the portion $\cdots \overset{N_i \quad N_{i+1}}{\wedge\!\wedge\!\!\times\!\!\wedge\!\wedge} \cdots$ of a bow diagram. We always choose a gauge with $\mathcal{A}_0 = 0$. The BF equations of motion (4.47) then imply, in particular, that $\mathcal{A}_3$ and $\sigma$ are independent of the $x^0$



$$\cdots \quad \overset{N_1}{\times} \quad \overset{N_2}{\times} \quad \overset{N_3}{\times} \quad \cdots \quad \overset{N_n}{\times}$$

Figure 5: The bow diagram for the D5 type boundary whose brane diagram is given in Fig. 4. Here $K = (K_1, \cdots, K_n)$ is an $n$-tuple of integers satisfying (4.57).

coordinate and therefore solving the equations of motion becomes a problem in one dimension. Thus, for a wavy line with label $N_i$ we have a $\mathfrak{gl}_{N_i}$ connection $\mathcal{D}_3 := \partial_3 + \mathcal{A}_3 \wedge$ and a covariantly constant (according to (4.48)) adjoint scalar $\sigma$. We assume that the D5 brane interface is located at $x^3 = 0$. We denote the fields to the left and to the right of the interface by $-$ and $+$ respectively, we thus denote the BF fields by $\mathcal{A}_3^\pm, \sigma^\pm$ and the 4d SYM fields by $A_3^\pm, X_1^\pm, X_2^\pm, X_3^\pm$. We consider two separate cases, $N_i \neq N_{i+1}$ and $N_i = N_{i+1}$.

- $N_i \neq N_{i+1}$. For concreteness let us assume $N_i > N_{i+1}$. The inequality $N_i > N_{i+1}$ means $N_{i+1}$ D3 branes pass through the D5 brane and the remaining $(N_i - N_{i+1})$ D3 branes end on the D5 brane from the left. We learn from [24] (see §3.4.4) that whenever D3 branes end on D5 branes, the adjoint scalars $\vec{X}$ from the 4d theory picks up poles compatible with the real Nahm's equations, and the scalars continue through the interface smoothly if the D3 branes extend past the D5 branes. More specifically, the real connection $A_3^+$ and the adjoint scalars $\vec{X}^+$ cross over smoothly from $x^3 > 0$ to $x^3 < 0$. But $\vec{X}^-$ picks up some poles at $x^3 = 0$ – near the interface it takes the form:[11]

$$\vec{X}^-(x^3) = \begin{pmatrix} \vec{X}^+(0) + O(x^3) & \vec{B} \\ \vec{C} & \frac{\vec{t}}{x^3} + O(1) \end{pmatrix}. \tag{4.58}$$

Here $\vec{B}$ and $\vec{C}$ are matrices of size $N_{i+1} \times (N_i - N_{i+1})$ and $(N_i - N_{i+1}) \times N_{i+1}$ respectively that are regular at $x^3 = 0$. The upper left corner of size $N_{i+1} \times N_{i+1}$ is regular and matches the value of $\vec{X}^+$ at the interface. In the lower right corner $\vec{t} = (t_1, t_2, t_3)$ are the images of a standard $\mathfrak{su}(2)$ basis under an irreducible homomorphism $\mathfrak{su}(2) \to \mathfrak{u}(N_i - N_{i+1})$. As usual, in the BF theory we only keep the constraints on $\sigma = X_2 - iX_3$. The aforementioned boundary condition then says that $\sigma^+$ is regular at the interface and $\sigma^-$ behaves near the interface as:

$$\sigma^-(x^3) = \begin{pmatrix} \sigma^+(0) + O(x^3) & B_2 - iB_3 \\ C_2 - iC_3 & \frac{t_2 - it_3}{x^3} + O(1) \end{pmatrix}, \tag{4.59}$$

where $t_2 - it_3 \in \mathfrak{gl}_{N_i - N_{i+1}}$ is a regular nilpotent element.

Boundary conditions for $N_i < N_{i+1}$ follows easily by swapping the superscript $+$ and $-$ on fields and changing $N_i - N_{i+1}$ to $N_{i+1} - N_i$.

- $N_i = N_{i+1}$. In this case we have the same gauge group on both sides of the D5 brane. Thus the D5 brane can be interpreted as a domain wall in a single gauge theory. In the 4d picture, we have $\mathcal{N} = 4$ U$(N_i)$ SYM and the open strings stretched between the stack of D3 branes and the D5 brane introduce a hypermultiplet, localized at the domain wall, transforming under the quaternionic representation $Z := T^*\mathbb{C}^{N_i}$ of the gauge group. The full 4d theory on $\mathbb{R}^4_{0123}$ can be viewed as a 3d theory with $\mathcal{N} = 4$ supersymmetry supported on $\mathbb{R}^3_{012}$ with fields valued in the infinite dimensional space of functions of $x^3$. The 4d fields $\vec{X}$ and $A_3$, as functions of $x^3$, become coordinates on an infinite dimensional hyper-Kähler target space. The scalars of the domain wall hypermultiplet parameterize

---

[11]D3 branes ending on D5s appear as monopoles in the three dimensional space transverse to the D3s and parallel to the D5s (parameterized by $\vec{X}$). It was shown in [49, 52] that solutions to Nahm's equations with the following boundary conditions coincide with such monopoles.

the finite dimensional hyper-Kähler target $Z$. From this point of view the gauge group is also infinite dimensional, consisting of $U(N_i)$ valued functions of $x^3$. This gauge group acts on the 4d fields, as well as on the fields localized at the domain wall. Only the restrictions of the group valued functions at the domain wall – which we take to be at $x^3 = 0$ – act on the localized fields. So we get a hyper-Kähler action of $U(N_i)$ on $Z$. The hyper-Kähler moment map of the infinite dimensional gauge group on the full target space is the following:[12]

$$\vec{\mu}(x^3) = D_3 \vec{X}(x^3) + \vec{X}(x^3) \times \vec{X}(x^3) + \delta(x^3)(\vec{X}^+(0) - \vec{X}^-(0) + \vec{\mu}^Z), \qquad (4.60)$$

where $\vec{\mu}^Z$ is the moment map of the $U(N_i)$ action on $Z$. Supersymmetric field configurations in the 4d theory are those on which the moment map vanishes. Away from the domain wall this simply means satisfying the standard Nahm's equation. The delta function supported at the domain wall implies that $\vec{X}$ is discontinuous at $x^3 = 0$ and jumps according to:

$$\vec{X}^+(0) - \vec{X}^-(0) + \vec{\mu}^Z = 0. \qquad (4.61)$$

Let us choose holomorphic coordinates on $Z$:

$$(I, J) \in \mathrm{Hom}(\mathbb{C}, \mathbb{C}^{N_i}) \oplus \mathrm{Hom}(\mathbb{C}^{N_i}, \mathbb{C}) = T^* \mathbb{C}^{N_i}, \qquad (4.62)$$

such that:

$$\vec{\mu}^Z = \left( \frac{1}{2}(II^\dagger - J^\dagger J), \mathrm{Re}\, IJ, -\mathrm{Im}\, IJ \right). \qquad (4.63)$$

After twisting and turning on omega deformation, the domain wall hypermultiplets localize to analytically continued topological quantum mechanics with target $Z$ coupled to the restriction of the gauge field of the BF theory at the domain wall [28, 29]. The kinetic term of the quantum mechanics simply takes the form $J \partial_0 I$. The coupling of the BF theory to this quantum mechanics via the term $J \mathcal{A}_0 I$ causes the BF field $\sigma$ to be discontinuous at the location of the defect and from (4.61) we get the following form of the discontinuity:

$$\sigma^+(0) - \sigma^-(0) + IJ = 0. \qquad (4.64)$$

This can also be seen as a Gauss law constraint in the coupled theory after gauging away $\mathcal{A}_0$.

Gauge symmetry is broken at a D5 interface for the gauge fields that satisfy Dirichlet boundary condition. For example, in case of $\cdots$  $\cdots$ with $N_i \geq N_{i+1}$ the $\mathfrak{u}(N_{i+1})$ part of the connection crosses over smoothly from one side to the other, but the components of the $\mathfrak{u}(N_i)$ connection orthogonal to the $\mathfrak{u}(N_{i+1})$ directions are set to zero at the domain wall by Dirichlet boundary condition. At the location of the D5 brane, gauge symmetry is reduced from $U(N_i)$ to $U(N_{i+1})$ going from left to right. Suppose $g^-(x_3)$ and $g^+(x_3)$ are gauge transformation matrices to the left and to the right of the D5 brane, then at $x^3 = 0$ they are required to satisfy:

$$g^-(0) = \begin{pmatrix} g^+(0) & 0 \\ 0 & \mathbb{1}_{N_i - N_{i+1}} \end{pmatrix}. \qquad (4.65)$$

Away from the interface the $\mathcal{A}_3$-covariant derivative $\mathcal{D}_3$ and the field $\sigma$ transform by the adjoint action as usual:

$$(\mathcal{D}_3, \sigma) \mapsto (g \mathcal{D}_3 g^{-1}, g \sigma g^{-1}). \qquad (4.66)$$

---

[12]If there are boundaries of the 4d theory away from the domain wall in either direction, there can be contribution to the moment map localized at those boundaries. We ignore this possibility here since we are now only interested in the boundary conditions coming from the domain wall.

Table 3: Directions wrapped by the D3 and NS5 branes from Fig. 6. We denote the D3 branes attached to the $j$th NS5 brane by D3$_j^\alpha$ for different values of $\alpha$. The fields $\vec{X}$ and $\vec{Y}$ of the 4d $\mathcal{N}=4$ theory living on the D3 branes consist of fluctuations of the D3 branes in $\mathbb{R}^3_{456}$ and $\mathbb{R}^3_{789}$ respectively, as in Table 2.

| | | $\mathbb{R}^2_{+\hbar}$ | | | | $\mathbb{R}^2_{-\hbar}$ | | | $C$ | |
| | 0 | 1 | 2 | 3 | 4 | 5 | 6 | 7 | 8 | 9 |
|---|---|---|---|---|---|---|---|---|---|---|
| Scalar Fields | | | | | $Y_1$ | $Y_2$ | $Y_3$ | $X_1$ | $X_2$ | $X_3$ |
| D3$_j^\alpha$ | × | × | × | × | | 0 | 0 | $r_{1,j}$ | $r_{2,j}$ | $r_{3,j}$ |
| NS5$_j$ | × | × | × | $\tau_j$ | × | × | × | $r_{1,j}$ | $r_{2,j}$ | $r_{3,j}$ |

This makes sense at the interface as well. In case $N_i = N_{i+1}$ and we have the additional fields $I$ and $J$ localized at the interface, they transform under the gauge transformation as:

$$(I,J) \mapsto (g(0)I, Jg(0)^{-1}). \tag{4.67}$$

### 4.2.2 NS5 Boundary

The most general NS5 type boundaries result from brane configurations such as Fig. 6. There

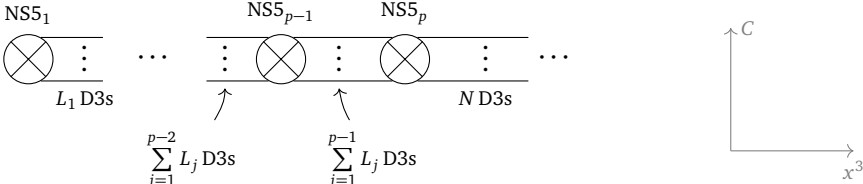

Figure 6: A generic NS5 type boundary determined by $\boldsymbol{L} = (L_1, \cdots, L_p)$ where $L_i$ is the linking number of the $i$th NS5 brane. The total number of D3 branes is $M := \sum_{j=1}^p L_j$.

are $p$ NS5 branes labeled NS5$_1$, ..., NS5$_p$ such that the linking number of NS5$_i$ is $L_i$. When there are no D5 branes to the left of any NS5 brane, as in Fig. 6, $L_i$ coincides with the number of D3 branes ending on NS5$_i$ from the right according to (2.2). The NS5 and the D3 brane world-volumes are as in Table 3. The $j$th NS5 brane is located at $x^3 = \tau_j$, $x^7 = r_{1,j}$, $x^8 = r_{2,j}$, $x^9 = r_{3,j}$. All the D3 branes are located at the center of the $\mathbb{R}^2_{-\hbar}$ plane to preserve the $U(1)_\hbar$ symmetry. We get a single line operator in 4d CS if all the D3 branes are coincident, in other words by taking the limit $\vec{r}_j := (r_{1,j}, r_{2,j}, r_{3,j}) \to \vec{r} := (r_1, r_2, r_3)$ for some $j$-independent point $\vec{r} \in \mathbb{R}^3_{789}$. In practice, we shall impose this limit only classically, adding $\mathcal{O}(\hbar)$ modifications as follows:

$$\vec{r} := \vec{r}_p, \qquad \vec{r}_j \to \vec{r}_{j+1} + \hbar \vec{\varrho}_j, \qquad j \in \{1, \cdots, p-1\}. \tag{4.68}$$

Here $\vec{\varrho}_i = (\varrho_{i,1}, \varrho_{i,2}, \varrho_{i,3}) \in \mathbb{R}^3_{789}$ are some arbitrary points. The terms proportional to $\hbar$ have no effect as far as classical brane geometry is concerned. These terms will affect our computations by deforming the moment map constraints coming from boundary conditions at NS5 domain walls. When we can describe the low energy theory of the D3 branes in terms of a 3d $\mathcal{N}=4$ quiver as in Fig. (3), the parameters $\vec{\varrho}_j$ correspond to the triplets of real FI parameter associated to the $j$th gauge node. In such cases deformations of the Higgs branch by FI parameters is a standard phenomenon. The locations of the D3 branes in the $x^4$ direction, i.e. the field $Y_1$, is not fixed at this boundary, rather it satisfies the Neumann boundary condition (3.20b).

As in the previous section, we treat this boundary by keeping the NS5 branes separated along the $x^3$-direction, assuming $\tau_1 < \tau_2 < \cdots < \tau_p$. Then each NS5 brane can be seen as a domain wall between two theories with possibly different gauge groups. In a bow diagram, we represent an NS5 brane by a hollow circle $\overset{\vec{r}}{\bigcirc}$ with its location in the $\mathbb{R}^3_{789}$ direction on top. As before, we draw the stack of D3 branes between two consecutive five-branes as a wavy line with a label showing the number of D3 branes. From Fig. 6 we see that for $i = 1, \cdots, p-1$ the number of D3 branes between $\mathrm{NS5}_j$ and $\mathrm{NS5}_{j+1}$ is:

$$M_j = \sum_{i=1}^{j} L_i \,. \tag{4.69}$$

To the right of of the last NS5 brane there are $M_p := M = \sum_{i=1}^{p} L_i$ D3 branes. The bow diagram corresponding to the brane configuration of Fig. 6 is in Fig. 7.

$$\overset{\vec{r}_1}{\bigcirc}\!\!\!\overset{M_1}{\wwww}\!\!\!\overset{\vec{r}_2}{\bigcirc}\overset{M_2}{\wwww}\cdots\overset{M_{p-1}}{\wwww}\overset{\vec{r}_p}{\bigcirc}\overset{M_p}{\wwww}\cdots$$

Figure 7: The bow diagram for the NS5 type boundary whose brane construction appears in Fig. 6. Here $\boldsymbol{L} = (L_1, \cdots, L_p)$ is a $p$-tuple of integers such that the $L_j$s are related to the $M_j$'s by (4.69). $\vec{r}_j$ is the location of $\mathrm{NS5}_j$ in $\mathbb{R}^3_{789}$ with the constraint (4.68) that $\vec{r}_{j+1} - \vec{r}_j = \hbar \vec{\varrho}_j$ for $j \in \{1, \cdots, p-1\}$.

Let us focus on the $j$th NS5 brane, i.e., the portion $\cdots \overset{M_{j-1}}{\wwww}\overset{\vec{r}_j}{\bigcirc}\overset{M_j}{\wwww}\cdots$ of a bow diagram. Before twisting and turning on $\Omega$-deformation the $j$th NS5 brane creates a domain wall between two 4d $\mathcal{N} = 4$ SYM theories with gauge groups $\mathrm{U}(M_{j-1})$ and $\mathrm{U}(M_j)$. Let us assume that the NS5 brane is located at $x^3 = 0$. We can write the half of the 4d space $\mathbb{R}^4_{0123}$ with $x^3 < 0$ as $\mathbb{R}^3_{012} \times L_-$ where $L_-$ is the negative half line. Similarly denote the other half of $\mathbb{R}^4_{0123}$ by $\mathbb{R}^3_{012} \times L_+$ where $L_+$ is the positive half line. We can think of the 4d $\mathcal{N} = 4$ SYM on $\mathbb{R}^4_{0123}$ as a 3d $\mathcal{N} = 4$ theory supported at $x^3 = 0$. The 3d theory has the infinite dimensional gauge group $\mathcal{G}^- \times \mathcal{G}^+ := \mathrm{Map}(L_-, \mathrm{U}(M_{j-1})) \times \mathrm{Map}(L_+, \mathrm{U}(M_j))$. The fields $\vec{X}$ and $A_3$, as functions of $L_\pm$, form two adjoint hypermultiplets of the 3d theory. The two hypermultiplets are acted on by the two factors of the gauge group respectively. Unlike a D5 brane, open strings can stretch across an NS5 brane, the open strings connecting the stacks of D3 branes to the left and to the right of $\mathrm{NS5}_j$ provide a bifundamental hypermultiplet to the 3d theory. This hypermultiplet transforms by the action of the restriction of the infinite dimensional gauge group to $x^3 = 0$. The restriction is simply $\mathrm{U}(M_{j-1}) \times \mathrm{U}(M_j)$ and the hypermultiplet transforms as $\widetilde{Z} := T^*\mathrm{Hom}(\mathbb{C}^{M_{j-1}}, \mathbb{C}^{M_j})$. The hyper-Kähler moment maps of the $\mathcal{G}_\pm$ actions on the hypermultiplets are:

$$\vec{\widetilde{\mu}}_\pm(x^3) = D_3 \vec{X}^\pm(x^3) + \vec{X}^\pm(x^3) \times \vec{X}^\pm(x^3) + \delta(x^3)\left(\pm\vec{X}^\pm(0) + \vec{\mu}_\pm^{\widetilde{Z}}\right). \tag{4.70}$$

Here $\vec{X}^\pm$ are functions on $L_\pm$ respectively, and $\mu_-^{\widetilde{Z}}$, $\mu_+^{\widetilde{Z}}$ are the moment maps for the actions of $\mathrm{U}(M_{j-1})$ and $\mathrm{U}(M_j)$ on $\widetilde{Z}$. By introducing holomorphic coordinates

$$(I, J) \in \mathrm{Hom}(\mathbb{C}^{M_{j-1}}, \mathbb{C}^{M_j}) \oplus \mathrm{Hom}(\mathbb{C}^{M_j}, \mathbb{C}^{M_{j-1}}) = \widetilde{Z}\,, \tag{4.71}$$

we can write these moment maps as:

$$\begin{aligned}
\vec{\mu}_+^{\widetilde{Z}} &= \left(\frac{1}{2}(J^\dagger J - II^\dagger), -\mathrm{Re}\, IJ, \mathrm{Im}\, IJ\right), \\
\vec{\mu}_-^{\widetilde{Z}} &= \left(\frac{1}{2}(JJ^\dagger - I^\dagger I), \mathrm{Re}\, JI, -\mathrm{Im}\, JI\right).
\end{aligned} \tag{4.72}$$

The boundary condition coming from the NS5 domain wall is simply that the terms in the moment maps localized at the domain wall vanish, with shifts by central elements depending on the location of $NS5_j$ in $\mathbb{R}^3_{789}$:

$$\vec{X}^+(0) + \vec{\mu}^{\widetilde{Z}}_+ = \vec{r}_j \, \mathbb{1}_{M_j}, \qquad \vec{X}^-(0) - \vec{\mu}^{\widetilde{Z}}_- = \vec{r}_j \, \mathbb{1}_{M_{j-1}}. \tag{4.73}$$

This becomes the following constraints on the BF fields:

$$\sigma^+(0) + IJ = r^{\mathbb{C}}_j \, \mathbb{1}_{M_j}, \qquad \sigma^-(0) + JI = r^{\mathbb{C}}_j \, \mathbb{1}_{M_{j-1}}, \tag{4.74}$$

where we have defined:

$$r^{\mathbb{C}}_j := r_{2,j} - i r_{3,j}. \tag{4.75}$$

$r^{\mathbb{C}}_j$ is the location of $NS5_j$ in the holomorphic $C$ direction. The real variable $r_{1,j}$ labeling the location of $NS5_j$ in the $x^7$ direction does not appear in the BF theory.

There is some ambiguity in the exact value of the central element in the equations (4.74). The field $\sigma^+$ appears in two equations, coming from the two NS5 interfaces on two sides of the wavy segment:

$$\cdots \overset{\displaystyle r^{\mathbb{C}}_j \quad M_j \quad r^{\mathbb{C}}_{j+1}}{\bigcirc\!\!\!\wedge\!\!\wedge\!\!\wedge\!\!\!\bigcirc} \cdots \, . \tag{4.76}$$

At each interface we have some localized bi-fundamental hypermultiplets. To distinguish the hypermultiplets from the left and the right interfaces, let us put a label on the fields as follows:

$$(I_k, J_k) \in \mathrm{Hom}(\mathbb{C}^{M_{k-1}}, \mathbb{C}^{M_k}) \oplus \mathrm{Hom}(\mathbb{C}^{M_k}, \mathbb{C}^{M_{k-1}}). \tag{4.77}$$

Then $(I_j, J_j)$ and $(I_{j+1}, J_{j+1})$ are the hypermultiplet fields at the left and the right interfaces respectively. If $NS5_j$ and $NS5_{j+1}$ are located at $x^3 = x_L$ and $x^3 = x_R$ respectively, then we get the following two constraints on $\sigma^+$:

$$\sigma^+(x_L) + I_j J_j = r^{\mathbb{C}}_j \, \mathbb{1}_{M_j}, \qquad \sigma^+(x_R) + J_{j+1} I_{j+1} = r^{\mathbb{C}}_{j+1} \, \mathbb{1}_{M_j}. \tag{4.78}$$

We can shift $\sigma^+$ by a constant amount: $\sigma^+ \to \sigma^+ + r^{\mathbb{C}}_{j+1} \mathbb{1}_{M_j}$, which changes these two equations to:

$$\sigma^+(x_L) + I_j J_j = \hbar \varrho^{\mathbb{C}}_j \, \mathbb{1}_{M_j}, \qquad \sigma^+(x_R) + J_{j+1} I_{j+1} = 0. \tag{4.79}$$

Here $\hbar \varrho^{\mathbb{C}}_j = r^{\mathbb{C}}_j - r^{\mathbb{C}}_{j+1}$ is the complex FI parameter for the abelian factor of the $U(M_j)$ gauge group associated with the $M_j$ D3 branes between the $j$th and the $(j+1)$st NS5 branes. We fix this ambiguity by adopting the convention that as in the second equation in (4.79), at a right NS5 interface there is no deformation of the moment map constraint, whereas as in the first equation in (4.79), at a left NS5 interface the moment map constraint is deformed by a central element given by the complex FI parameter.

Gauge symmetry is not broken at an NS5 interface. If $g^-$ and $g^+$ are the gauge transformation matrices on the left and the right of an NS5 interface located at $x^3 = 0$, then the bifundamental fields $I$ and $J$ at the interface transform under the gauge transformation as:

$$(I, J) \mapsto (g^+(0) I g^-(0)^{-1}, g^-(0) J g^+(0)^{-1}). \tag{4.80}$$

## 4.3 Phase Spaces of BF Theories with Boundaries

### 4.3.1 BF Theories and Cherkis Bows

Putting these all together, we create a 2d BF theory with $GL_N$ gauge group as follows. We suspend $N$ D3 branes between $n$ D5 and $p$ NS5 branes. The boundary configurations

are described by the linking numbers $\boldsymbol{K} = (K_1, \cdots, K_n)$ and $\boldsymbol{L} = (L_1, \cdots, L_p)$ satisfying $\sum_{i=1}^{n} K_i = \sum_{i=j}^{p} L_j = N$. When all the NS5s are positioned to the left of all the D5s, $K_i$ and $L_j$ are the net numbers of D3 branes ending on $\text{D5}_i$ and $\text{NS5}_j$ from the left and from the right respectively (cf. (2.2)). Additionally, we need to keep track of the locations $r_j^{\mathbb{C}}$ of the NS5 branes in the $C$ direction. The theories will depend only on the differences between positions, namely (cf. (4.68), (2.3), and (4.75)):

$$\hbar\varrho_j^{\mathbb{C}} = r_j^{\mathbb{C}} - r_{j+1}^{\mathbb{C}}, \qquad j \in \{1, \cdots, p-1\}. \tag{4.81}$$

Also define:

$$z := r_p^{\mathbb{C}}. \tag{4.82}$$

This is the spectral parameter associated to the line operator. Since this is coordinate dependent, the field theories, or the phase spaces associated to the line operators are independent of it. The BF theory constructed from such brane configurations is denoted by $T_{\varrho}^{\text{BF}}(\boldsymbol{K}, \boldsymbol{L})$ where $\boldsymbol{\varrho} := (\varrho_1^{\mathbb{C}}, \cdots, \varrho_{p-1}^{\mathbb{C}})$.

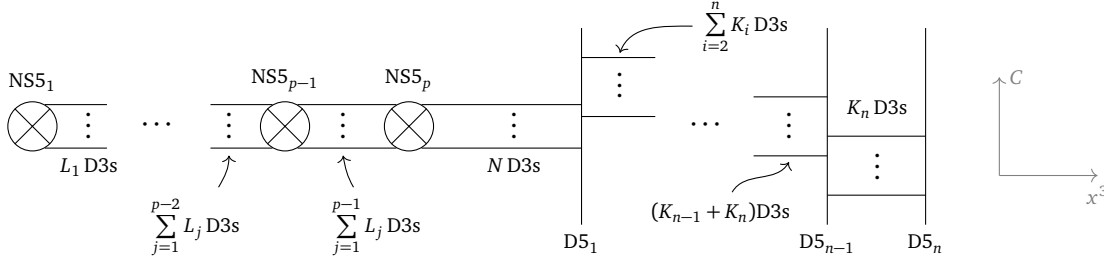

Figure 8: Brane configuration for the $\text{GL}_N$ BF theory $T_{\varrho}^{\text{BF}}(\boldsymbol{K}, \boldsymbol{L})$ on an interval labeled by the linking numbers $\boldsymbol{K}$ and $\boldsymbol{L}$. This figure is a duplicate of Fig. 1.

The worldvolume theory of the D3 branes is the 4d $\mathcal{N} = 4$ U($N$) SYM theory with two 1/2-BPS boundaries. Turning on $\Omega$-deformation localizes the 4d theory to the 2d BF theory with an adjoint scalar $\sigma$ and a gauge field $\mathcal{A}$. We associate the following bow diagram – denoted by $\text{Bow}_{\varrho}(\boldsymbol{K}, \boldsymbol{L})$ – to this BF theory:

Figure 9: The bow diagram $\text{Bow}_{\varrho}(\boldsymbol{K}, \boldsymbol{L})$ defined by $\boldsymbol{K} = (K_1, \cdots K_n)$ and $\boldsymbol{L} = (L_1, \cdots L_p)$ satisfying $N_i = \sum_{j=i}^{n} K_j$ and $M_i = \sum_{j=1}^{i} L_j$ with $M_p = N_1 = N$. $\boldsymbol{\varrho} = (\varrho_1^{\mathbb{C}}, \cdots, \varrho_{p-1}^{\mathbb{C}})$ contains deformation parameters where $\hbar\varrho_j^{\mathbb{C}} = r_j^{\mathbb{C}} - r_{j+1}^{\mathbb{C}}$.

A wavy line segment with the label $m$ corresponds to a $\text{GL}_m$ BF theory between two interfaces. We gauge away the component of the gauge field parallel to the interfaces. Then each line segment represents the equation of motion $\mathcal{D}_3\sigma = 0$ where $\mathcal{D}_3$ is the gauge covariant derivative in the direction normal to the interfaces. At the D5 and NS5 interfaces boundary conditions are imposed as described in the previous sections. The space of solutions to the equation of motion subjected to the boundary conditions modulo gauge transformations is called the Cherkis Bow Variety as defined in [6,7], which we denote by $\mathcal{M}_{\varrho}^{\text{bow}}(\boldsymbol{K}, \boldsymbol{L})$.

$$\mathcal{P}_{\varrho}^{\text{BF}}(\boldsymbol{K}, \boldsymbol{L}) := \text{Phase space of } T_{\varrho}^{\text{BF}}(\boldsymbol{K}, \boldsymbol{L}) = \mathcal{M}_{\varrho}^{\text{bow}}(\boldsymbol{K}, \boldsymbol{L}). \tag{4.83}$$

### 4.3.2 BF Theories and Branches of Vacua of 3d $\mathcal{N} = 4$ Theories

We have constructed the BF theory on an interval by applying $\Omega$-deformation to a 4d theory on an interval. By the topological symmetry of the 4d theory, we can in principle consider the limit where the interval shrinks to zero and we have an $\Omega$-deformed 3d theory which localizes to a TQM. Of course, the phase space remains invariant under this scaling and the phase space of the TQM is the same as the phase space of the BF theory. The problem in implementing this in general is that we do not have a concrete description of the effective 3d theory for arbitrary linking numbers $K$ and $L$ of the five-branes in the brane construction such as in Fig. 8. Regardless, in §2 we gave the name $T^{\vee}[\mathrm{U}(N)]^L_K$ to this 3d theory which has $\mathcal{N} = 4$ supersymmetry. We know on general grounds [28, 29] that a 3d $\mathcal{N} = 4$ theory reduces, upon the B-type $\Omega$-deformation, to a TQM whose phase space is the Higgs branch of the 3d theory. In other words,

$$\mathcal{P}^{\mathrm{BF}}_{\varrho}(K, L) = \mathcal{M}_H(T^{\vee}_{\varrho}[\mathrm{U}(N)]^L_K). \tag{4.84}$$

Using mirror symmetry, and denoting the mirror of $T^{\vee}_{\varrho}[\mathrm{U}(N)]^L_K$ by $T_{\varrho}[\mathrm{U}(N)]^L_K$ we get the same BF phase space as a Coulomb branch:

$$\mathcal{P}^{\mathrm{BF}}_{\varrho}(K, L) = \mathcal{M}_C(T_{\varrho}[\mathrm{U}(N)]^L_K) := \text{Coulomb branch of } T_{\varrho}[\mathrm{U}(N)]^L_K. \tag{4.85}$$

Note that mirror symmetry exchanges FI parameters with twisted masses [8] and so in the mirror theory $T_{\varrho}[\mathrm{U}(N)]^L_K$, the parameters $\hbar\varrho$ are twisted masses. Mirror symmetry is realized in type IIB string theory as S-duality [19] which changes D5 branes to NS5s, NS5s to D5s, and leaves the D3 branes as D3s. Once again, for generic linking numbers of the five-branes, neither $T^{\vee}_{\varrho}[\mathrm{U}(N)]^L_K$ nor $T_{\varrho}[\mathrm{U}(N)]^L_K$ has a quiver description and we can not give a more concrete description of their branches of vacua than saying that they are the Cherkis bow varieties we have computed in the last subsection.

To make contact with existing results about vacuum branches, we now specialize to brane configurations where $T_{\varrho}[\mathrm{U}(N)]^L_K$ has a quiver description. The requirements are similar to (2.8) of §2.2. In that section we looked at example where $T^{\vee}_{\varrho}[\mathrm{U}(N)]^L_K$ has a quiver description, now we need the "mirror" constraints on the linking numbers, namely:[13]

$$0 < L_j < n, \qquad 1 \le j \le p,$$

$$v_i := -\sum_{j=1}^{i} K_j + \sum_{j=1}^{n-1} \min(i,j) w_j \ge 0, \qquad 1 \le j < n, \tag{4.86}$$

$$\text{where,} \quad w_i := \#\{L_j \mid L_j = i\}.$$

With these constraints, we can bring the NS5 branes between D5 branes such that there are equal numbers of D3 branes on both sides of every NS5 brane. Then by applying S-duality we find the 3d $\mathcal{N} = 4$ theory $T_{\varrho}[\mathrm{U}(N)]^L_K$ defined by the quiver in Fig. 10. This is a gauge theory with the gauge group $\prod_{i=1}^{n-1} \mathrm{U}(v_i)$ and flavor group $\prod_{i=1}^{n-1} \mathrm{SU}(w_i)$. An edge between two nodes corresponding to two groups $\mathrm{U}(v)$ and $\mathrm{U}(w)$ represents a bifundamental hypermultiplet transforming under the representation $T^*\mathrm{Hom}(\mathbb{C}^v, \mathbb{C}^w)$ of $\mathrm{U}(v) \times \mathrm{U}(w)$. It was proved in [7] that the Coulomb branch of this theory is a bow variety:

$$\mathcal{M}_C(T_{\varrho}[\mathrm{U}(N)]^L_K) = \mathcal{M}^{\mathrm{bow}}_{\varrho}(K, L). \tag{4.87}$$

Here $\mathcal{M}^{\mathrm{bow}}_{\varrho}(K, L)$ is precisely the bow variety associated to the bow diagram of Fig. 9. The relation (4.85) between BF phase spaces and Coulomb branches now implies our earlier claim (4.83). Of course, our claim is that (4.83) holds true for more general linking numbers, even when the 3d theories involved have no quiver descriptions.

---

[13]The bow varieties constructed with these constraints are given by what are called balanced dimension vectors in [7].

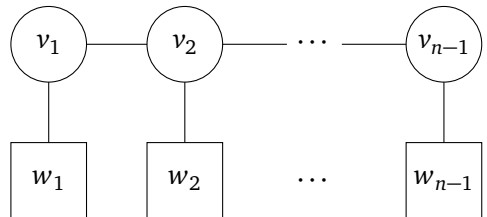

Figure 10: Quiver for the 3d $\mathcal{N} = 4$ theory $T_\varrho[\mathrm{U}(N)]_K^L$ with $N = \sum_{i=1}^n K_i = \sum_{j=1}^p L_j$. The ranks of the gauge and flavor groups are defined form $K$ and $L$ via (4.86). $\varrho = (\varrho_1^\mathbb{C}, \cdots, \varrho_{p-1}^\mathbb{C})$ where $\hbar \varrho_j^\mathbb{C}$s are complex twisted masses for the flavor symmetry. The mirror of this theory is denoted by $T_\varrho^\vee[\mathrm{U}(N)]_K^L$, if this also admits a quiver description then the quiver is given by Fig. 3.

## 5 Line Operators in 4d Chern-Simons Theory

The line operator $\mathbb{L}_\varrho(K, L)$, created by the brane configuration of Fig. 8, appears in 4d CS theory as a coupled TQM whose target space is the phase space of the BF theory $T_\varrho^{\mathrm{BF}}(K, L)$. Thus,

$$\text{Phase space of } \mathbb{L}_\varrho(K, L) = \mathcal{P}_\varrho^{\mathrm{BF}}(K, L) = \mathcal{M}_\varrho^{\mathrm{bow}}(K, L). \tag{5.1}$$

The equality between BF phase spaces and bow varieties is from (4.83). The TQM quantizes the algebra of functions on this phase space. So this algebra $\mathcal{A}_\varrho(K, L)$, which couples to the line operator, can be characterized as the deformation quantization of functions on Bow varieties:

$$\mathcal{A}_\varrho(K, L) = \text{Deformation quantization of } \mathbb{C}[\mathcal{M}_\varrho^{\mathrm{bow}}(K, L)]. \tag{5.2}$$

This algebra will show up in the study of integrable spin chains as follows. Consider a $\mathfrak{gl}_n$ spin chain of length $L$ consisting of spins $\mathcal{R}_1(z_1), \cdots, \mathcal{R}_L(z_L)$. Here $\mathcal{R}_i(z_i)$ is the evaluation module of the Yangian of $\mathfrak{gl}_n$ associated with the $\mathfrak{gl}_n$ module $\mathcal{R}_i$ and spectral parameter $z_i$. The Hilbert space of the spin chain is the tensor product $\mathcal{H} := \bigotimes_{i=1}^L \mathcal{R}_i(z_i)$. Following [13], this spin chain can be created in 4d $\mathrm{GL}_n$ CS theory by putting $L$ parallel Wilson lines carrying the representations $\mathcal{R}_i$. The Wilson lines are along the topological plane of 4d CS and have fixed locations $z_i$ in the holomorphic $\mathbb{C}$ direction. Viewing these Wilson lines as vertical we can introduce $\mathbb{L}_\varrho^{z_0}(K, L)$ as a horizontal line operator, as in Fig. 11. Here $z_0$ is simply the location of the operator in the $\mathbb{C}$ direction. This horizontal line operator creates a monodromy

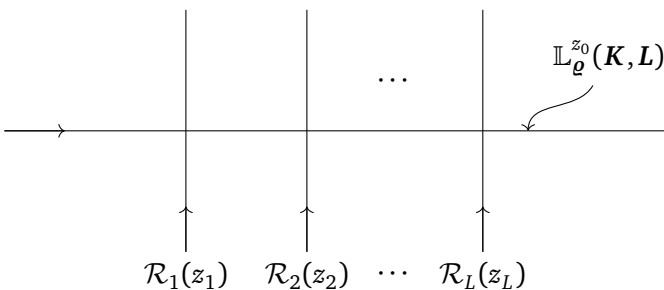

Figure 11: Schematic diagram of a $\mathfrak{gl}_n$ spin chain with Hilbert space $\mathcal{H} = \bigotimes_{i=1}^L \mathcal{R}_i(z_i)$, realized in 4d $\mathrm{GL}_n$ CS theory as an arrangement of vertical Wilson lines. $\mathbb{L}_\varrho^{z_0}(K, L)$ has been introduced as a horizontal line operator to create the monodromy matrix $\mathcal{L}_\varrho^{z_0}(K, L | \mathcal{H})$.

matrix which we denote by $\mathcal{L}_{\varrho}^{z_0}(K, L|\mathcal{H})$. This monodromy matrix acts on an extended space $V \otimes \mathcal{H}$ where the auxiliary space $V$ is some module for $\mathcal{A}_{\varrho}(K, L)$. Without studying geometric quantization of $\mathcal{M}_{\varrho}^{\text{bow}}(K, L)$ we can not say exactly which module to assign to this line operator. What we can say at this stage is which operator algebra the monodromy matrix belongs to:

$$\mathcal{L}_{\varrho}^{z_0}(K, L|\mathcal{H}) \in \mathcal{A}_{\varrho}(K, L) \otimes \text{End}(\mathcal{H}). \tag{5.3}$$

In other words, the monodromy matrix can be seen as a matrix that acts on $\mathcal{H}$ with entries in the algebra $\mathcal{A}_{\varrho}(K, L)$. Taking a partial trace of the monodromy matrix over the $\mathcal{A}_{\varrho}(K, L)$-module produces a transfer matrix. Some closely related traces were studied recently in [45,53]. Note that while the monodromy matrix itself will generally depend on the spectral parameter $z_0$ of the horizontal line, the algebra $\mathcal{A}_{\varrho}(K, L)$ is independent of it.[14]

Integrability of the spin chain means that the monodromy matrices satisfy the RTT relations given the R-matrix. In [21] the authors explicitly constructed a class of monodromy matrices that are linear in $z_0$ for the Hilbert space $\mathcal{H} = \mathcal{R}_1(z_1) = \mathbb{C}^n$ being the fundamental representation of $\mathfrak{gl}_n$. Examples of these monodromy matrices include the T, Q, and the L-operators corresponding to elements of $U(\mathfrak{gl}_n) \otimes \text{End}(\mathbb{C}^n)$, $\text{Weyl}^{\otimes k(n-k)} \otimes \text{End}(\mathbb{C}^n)$, and $U(\mathfrak{gl}_k) \otimes \text{Weyl}^{\otimes k(n-k)} \otimes \text{End}(\mathbb{C}^n)$ respectively. Below we describe the bow varieties and the quivers defining the line operators that create monodromy matrices associated with the algebras $U(\mathfrak{gl}_n)$, $\text{Weyl}^{\otimes k(n-k)}$, and $U(\mathfrak{gl}_k) \otimes \text{Weyl}^{\otimes k(n-k)}$. We conjecture that they are in fact the T, Q, and L-operators from the literature.

By some abuse of notation, in the following we use the terms T, Q, and L-operators to refer to the horizontal line operator from Fig. 11 for certain $K$ and $L$ and not the corresponding element of $\mathcal{A}_{\varrho}(K, L) \otimes \text{End}(\mathcal{H})$.

## 5.1 Example: The T-Operators (Wilson Lines)

A basic example of line operators in 4d $GL_n$ CS theory is the T-operator. In our notation (1.2) a T-operator $\mathbb{L}_{\varrho}^{z}(K, L)$ is characterized by having

$$K = L = (\overbrace{1, \cdots, 1}^{n}). \tag{5.4}$$

Given the mass parameters[15] $\varrho = (\varrho_1^{\mathbb{C}}, \cdots, \varrho_{n-1}^{\mathbb{C}})$, the phase space of this operator is the Coulomb branch of the 3d $\mathcal{N} = 4$ theory $T_{\varrho}[U(N)]_{(1,\cdots,1)}^{(1,\cdots,1)}$. This is a much studied theory in the 3d $\mathcal{N} = 4$ literature and it is often denoted simply as $T_{\varrho}[U(n)]$, which we adopt in this section. We shall prove at the end of this section that

**Proposition 5.1.** The T-operator, i.e., $\mathbb{L}_{\varrho}^{z}(K, L)$ for $K$ and $L$ given by (5.4) carries a $\mathfrak{gl}_n$ Verma module. Up to the action of the Weyl group, the Verma module has the highest weight $\lambda - \rho$. The weight $\lambda$ is determined by its Dynkin labels $(\varrho_1^{\mathbb{C}}, \cdots, \varrho_{n-1}^{\mathbb{C}})$, and the *Weyl vector* $\rho = \frac{1}{2} \sum_{\alpha \in \Delta^+} \alpha = (\frac{n-1}{2}, \frac{n-3}{2}, \cdots, -\frac{n-1}{2})$ is the half-sum of the positive roots.

To prove this proposition we shall compute the values of the Casimirs of $\mathfrak{gl}_n$ take in these modules. This gives us a Verma module assuming the representation is of the highest weight type. More general representations are possible but not considered in this paper.

---

[14]If we treat the spectral parameter $z$ as a formal variable, as opposed to a complex number, then there may be a factor of $\mathbb{C}[z]$ in the algebra, for example see §5.1.

[15]Mass parameters can be either twisted masses or FI parameters depending on the duality frame.

The theory $T_\varrho[\mathrm{U}(n)]$ can be described by the following quiver.

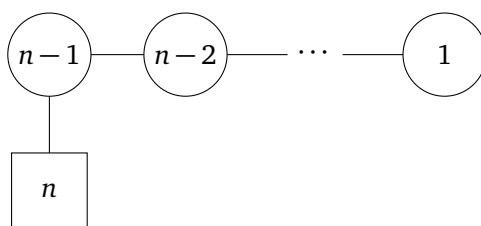

Figure 12: The quiver for the 3d $\mathcal{N} = 4$ theory $T_\varrho[\mathrm{U}(n)]$, whose Coulomb branch is the phase space for the T-operator.

It is well-known that the Coulomb branch of $T_\varrho[\mathrm{U}(n)]$ is a deformation of the nilpotent cone $\mathcal{N}_{\mathfrak{gl}_n}$ [11]. If we turn on all complex twisted masses and treat them as undetermined parameters, then the Coulomb branch is $\mathfrak{gl}_n^*$,[16] and it quantizes to the universal enveloping algebra $U_\hbar(\mathfrak{gl}_n)$ [56, Example 6.2]. Incorporate the mass parameters $\varrho$ and the spectral parameter $z$ into the following complex numbers:

$$r_i := z + \hbar \sum_{j=i}^{n-1} \varrho_j^{\mathbb{C}}, \qquad 1 \le i \le n. \tag{5.5}$$

Then the Coulomb branch can be equivalently described by the following bow diagram.

$$
\begin{array}{c}
\overset{r_1}{\phantom{.}} \; \underset{1}{\bigcirc} \!\!\sim\!\! \overset{r_2}{\phantom{.}} \underset{2}{\bigcirc} \!\!\sim\!\! \cdots \!\!\sim\!\! \underset{n-1}{\phantom{.}} \overset{r_n}{\phantom{.}} \underset{n}{\bigcirc} \!\!\sim\!\! \times^{n-1} \!\!\sim\!\! \cdots \!\!\sim\!\! {}^{2}\!\!\times\!\! {}^{1}\!\!\times
\end{array}
$$

Figure 13: Bow diagram whose associated bow variety is the Coulomb branch of $T_\varrho[\mathrm{U}(n)]$ from Fig. 12.

The theory $T_\varrho[\mathrm{U}(n)]$ is self mirror and so the dual theory $T_\varrho^\vee[\mathrm{U}(n)]$ is also described by the same quiver as in Fig. 12. Let $\mathcal{A}_n$ be the quantized Higgs branch algebra of $T_\varrho^\vee[\mathrm{U}(n)]$, then $\mathcal{A}_n$ is the quantum Hamiltonian reduction of $\mathbb{C}[r_n] \otimes \bigotimes_{k=1}^{n-1} \mathrm{Weyl}_\hbar^{\otimes k(k+1)}$ with respect to the action of $\bigoplus_{k=1}^{n-1} \mathfrak{gl}_k$. Let us introduce the variables:

$$I_k \in \mathrm{Hom}(\mathbb{C}^{k-1}, \mathbb{C}^k), \quad J_k \in \mathrm{Hom}(\mathbb{C}^k, \mathbb{C}^{k-1}), \qquad 1 \le k \le n-1, \tag{5.6}$$

satisfying the Weyl algebra commutation relations:

$$[J_{k,j}^\alpha, I_{k,\beta}^i] = \hbar \delta_\beta^\alpha \delta_j^i, \qquad 1 \le i,j \le k, \quad 1 \le \alpha,\beta \le k-1. \tag{5.7}$$

We shall use Weyl ordering to promote functions of classical variables into functions of operators:

$$(PQ)_W := \frac{1}{2}(PQ + QP). \tag{5.8}$$

Now we can write down the quantum moment map equations for the Hamiltonian reduction:

$$(I_{k,\alpha}^i J_{k,j}^\alpha)_W - (I_{k+1,j}^a J_{k+1,a}^i)_W + \hbar \varrho_k^{\mathbb{C}} \delta_j^i = I_{k,\alpha}^i J_{k,j}^\alpha - I_{k+1,j}^a J_{k+1,a}^i - \hbar \delta_j^i + \hbar \varrho_k^{\mathbb{C}} \delta_j^i = 0. \tag{5.9}$$

[16]Using [54, Theorem 7.6.1] and [55, Theorem 2.11].

**Lemma 5.2.** Treat the $r_i$'s from (5.5) as formal variables, then $\mathcal{A}_n$ is isomorphic to the centrally extended universal enveloping algebra

$$U_\hbar(\mathfrak{gl}_n) \otimes_{U_\hbar(\mathfrak{gl}_n)^{\mathrm{GL}_n}} \mathbb{C}[r_1, \cdots, r_n, \hbar], \tag{5.10}$$

where the map $U_\hbar(\mathfrak{gl}_n)^{\mathrm{GL}_n} \to \mathbb{C}[r_1, \cdots, r_n, \hbar]$ is given by

$$C(X, u) = \prod_{k=1}^{n} \left( u - r_k - \frac{n-1}{2}\hbar \right). \tag{5.11}$$

Here $C(X, u)$ is the Capelli determinant[17] of the generators $X^i_j$ of $U_\hbar(\mathfrak{gl}_n)$ with standard commutation relations $[X^i_j, X^l_m] = \hbar \delta^l_j X^i_m - \hbar \delta^i_m X^l_j$. Moreover, the explicit isomorphism is given by

$$X^i_j \mapsto (I^i_{n,\alpha} J^\alpha_{n,j})_W + r_n \delta^i_j = I^i_{n,\alpha} J^\alpha_{n,j} + \frac{\hbar}{2}(n-1) + r_n \delta^i_j. \tag{5.12}$$

*Proof.* We prove the lemma by induction on $n$. The case $n = 1$ is obvious, and we prove the induction step as follows. By induction, $\mathcal{A}_{n-1}$ is isomorphic to $U_\hbar(\mathfrak{gl}_{n-1}) \otimes_{U_\hbar(\mathfrak{gl}_{n-1})^{\mathrm{GL}_{n-1}}} \mathbb{C}[r_1, \cdots, r_{n-1}, \hbar]$, and $U_\hbar(\mathfrak{gl}_{n-1})^{\mathrm{GL}_{n-1}} \to \mathbb{C}[r_1, \cdots, r_{n-1}, \hbar]$ is given by $C(Y, u) = \prod_{k=1}^{n-1}\left(u - r_k - \frac{n-2}{2}\hbar\right)$, where $C(Y, u)$ is the Capelli determinant of generators $Y^a_b$ of $U_\hbar(\mathfrak{gl}_{n-1})$, and the explicit isomorphism is given by $Y^a_b \mapsto I^a_{n-1,\alpha} J^\alpha_{n-1,b} + \frac{\hbar}{2}(n-2) + r_{n-1}\delta^a_b$. Then $\mathcal{A}_n$ is the quantum Hamiltonian reduction of $\mathbb{C}[r_n] \otimes \mathrm{Weyl}_\hbar^{\otimes n(n-1)} \otimes \mathcal{A}_{n-1}$ with respect to the $\mathfrak{gl}_{n-1}$ action and moment map equation

$$Y^a_b - I^i_{n,b} J^a_{n,i} - \left(r_n + \frac{n}{2}\hbar\right)\delta^a_b = 0. \tag{5.13}$$

Now recall the notation of section 6.1 of [56], we can set $B_+ := Y - (r_n + \frac{n-2}{2}\hbar)\mathbb{1}_{n-1}$, $\psi = J$, $\overline{\psi} = I$ and then quotient by the right ideal generated by $B_-$ (which turns out to be a two-sided ideal), and then we obtain an algebra homomorphism $\mathbb{C}_\hbar[\mathcal{M}(n-1, n)] \to \mathcal{A}_n$. Under this map, we have

$$\begin{aligned}
\mathrm{qdet}\left(\mathbb{1}_n - I\frac{1}{u - B_-}J\right) &\mapsto \mathrm{qdet}\left(\mathbb{1}_n - \frac{\widetilde{X}}{u}\right) \qquad [\text{we defined, } \widetilde{X} := X - \left(r_n + \frac{\hbar}{2}(n-1)\right)\mathbb{1}_n] \\
&= \sum_{\sigma \in \mathfrak{S}_n} \mathrm{sgn}(\sigma)\left(\delta^{\sigma(1)}_1 - \frac{\widetilde{X}^{\sigma(1)}_1}{u - \frac{n-1}{2}\hbar}\right) \cdots \left(\delta^{\sigma(n)}_n - \frac{\widetilde{X}^{\sigma(n)}_n}{u + \frac{n-1}{2}\hbar}\right) \\
&= \frac{C(X, u + r_n + (n-1)\hbar)}{(u + \frac{n-1}{2}\hbar)\cdots(u - \frac{n-1}{2}\hbar)}.
\end{aligned} \tag{5.14}$$

On the other hand, [56, Proposition 6.2] implies that[18]

$$\mathrm{qdet}\left(\mathbb{1}_n - I\frac{1}{u - B_-}J\right) = \frac{C(B_+, u + \frac{n-1}{2}\hbar)}{(u + \frac{n-3}{2}\hbar)(u + \frac{n-5}{2}\hbar)\cdots(u - \frac{n-1}{2}\hbar)}$$

---

[17]The Capelli determinant $C(B, u)$ of operators $B^i_j$ satisfying $\mathfrak{gl}_n$ commutation relations $[B^i_j, B^l_m] = \hbar\delta^l_j B^i_m - \hbar\delta^i_m B^l_j$ is defined as $C(B, u) = \sum_{\sigma \in \mathfrak{S}_n} \mathrm{sgn}(\sigma)(u - (n-1)\hbar - B)^{\sigma(1)}_1 \cdots (u - B)^{\sigma(n)}_n$.

[18]We use the identity $\mathrm{qdet}\left(\mathbb{1}_n - I\frac{1}{u-B_-}J\right)\mathrm{qdet}\left(\mathbb{1}_n + I\frac{1}{u-B_+}J\right) = 1$ [57] to derive the first equality from [56, Proposition 6.2].

$$= \frac{C(Y, u + r_n + (n - \frac{3}{2})\hbar)}{(u + \frac{n-3}{2}\hbar) \cdots (u - \frac{n-1}{2}\hbar)} \,. \tag{5.15}$$

Combining the above two equations we see that

$$C(X, u) = \left(u - r_n - \frac{n-1}{2}\hbar\right) C\left(Y, u - \frac{\hbar}{2}\right),$$

and by the induction hypothesis $C\left(Y, u - \frac{\hbar}{2}\right) = \prod_{k=1}^{n-1} \left(u - r_k - \frac{n-1}{2}\hbar\right)$, thus

$$C(X, u) = \prod_{k=1}^{n} \left(u - r_k - \frac{n-1}{2}\hbar\right) \,. \tag{5.16}$$

Since $\mathcal{A}_n$ is generated by $X_j^i$ and $r_1, \cdots, r_n$ as a $\mathbb{C}[\hbar]$ algebra, we obtain a surjective map

$$U_\hbar(\mathfrak{gl}_n) \otimes_{U_\hbar(\mathfrak{gl}_n)^{\mathrm{GL}_n}} \mathbb{C}[r_1, \cdots, r_n, \hbar] \twoheadrightarrow \mathcal{A}_n.$$

Since modulo $\hbar$, the former algebra becomes the ring of functions on the variety $\mathfrak{gl}_n^* \times_{\mathfrak{gl}_n^*/\mathrm{GL}_n} \mathfrak{a}_n^*$ where $\mathfrak{a}_n$ is the Cartan subalgebra, and the map $\mathfrak{a}_n^* \to \mathfrak{gl}_n^*/\mathrm{GL}_n$ is identified with the quotient by the Weyl group $\mathfrak{a}_n^* \to \mathfrak{a}_n^*/\mathfrak{S}_n \cong \mathfrak{gl}_n^*/\mathrm{GL}_n$. It is known that $\mathfrak{gl}_n^* \times_{\mathfrak{gl}_n^*/\mathrm{GL}_n} \mathfrak{a}_n^*$ is a normal variety of dimension $n^2$, and $\mathcal{A}_n/(\hbar)$ is the ring of function on a (deformation of) Nakajima quiver variety of dimension $n^2$,[19] thus the surjective map $U_\hbar(\mathfrak{gl}_n) \otimes_{U_\hbar(\mathfrak{gl}_n)^{\mathrm{GL}_n}} \mathbb{C}[r_1, \cdots, r_n, \hbar] \twoheadrightarrow \mathcal{A}_n$ is an isomorphism modulo $\hbar$. Since both sides of the map are flat over $\mathbb{C}[\hbar]$, the map must be injective as well, thus $\mathcal{A}_n$ is isomorphic to $U_\hbar(\mathfrak{gl}_n) \otimes_{U_\hbar(\mathfrak{gl}_n)^{\mathrm{GL}_n}} \mathbb{C}[r_1, \cdots, r_n, \hbar]$. □

**Remark 5.3** (Comparing the Higgs and Coulomb Branches). The reader might notice the difference between the Coulomb branch algebra of $T_\varrho[\mathrm{U}(n)]$, which is $U_\hbar(\mathfrak{gl}_n)$, and the Higgs branch algebra of $T_\varrho^\vee[\mathrm{U}(n)]$ which is $U_\hbar(\mathfrak{gl}_n) \otimes_{U_\hbar(\mathfrak{gl}_n)^{\mathrm{GL}_n}} \mathbb{C}[r_1, \cdots, r_n, \hbar]$. This difference is actually superfluous, since mass parameters in our definition of Coulomb branch are symmetrized, in the sense that we only take symmetric polynomials in the mass parameters, and they form the $\mathrm{GL}_n$-invariant part $U_\hbar(\mathfrak{gl}_n)^{\mathrm{GL}_n} \cong \mathbb{C}[r_1, \cdots, r_n, \hbar]^{S_n}$ of the Coulomb branch algebra, where $S_n$ permutes $r_1, \cdots, r_n$. On the other hand, FI parameters in our definition of Higgs branch are not symmetrized. If we extend the the Coulomb side by adding non-symmetric polynomials of mass parameters, we get $U_\hbar(\mathfrak{gl}_n) \otimes_{U_\hbar(\mathfrak{gl}_n)^{\mathrm{GL}_n}} \mathbb{C}[r_1, \cdots, r_n, \hbar]$, which is isomorphic to the Higgs branch algebra of $T_\varrho^\vee[\mathrm{U}(n)]$.

*Proof of Proposition 5.1.* In practice, the $r_i$'s from (5.5) are locations of the NS5 branes in the holomorphic $\mathbb{C}$ direction and so we should treat them as complex numbers instead of formal variables. If we evaluate the algebra $\mathcal{A}_n$ from Lemma 5.2 at $r_k = \lambda_k \hbar, \lambda_k \in \mathbb{C}$, then we get the central quotient algebra $U_\hbar(\mathfrak{gl}_n)/\left(C(X, u) - \prod_{k=1}^{n}(u - (\lambda_k + \frac{n-1}{2})\hbar)\right)$, which acts on the Verma module with the highest weight (with some choice of ordering for the $\mathfrak{gl}_n$ fundamental weights)

$$(\lambda_1, \cdots, \lambda_n) - \rho \,. \tag{5.17}$$

This follows from the fact that the Capelli determinant acts on the ground state $|\lambda\rangle$ as

$$C(X, u)|\lambda\rangle = \prod_{k=1}^{n} \left(u - (n-k)\hbar - X_k^k\right)|\lambda\rangle \,, \tag{5.18}$$

---

[19]Let $\mathfrak{h}$ be the Cartan subalgebra of the Lie algebra $\mathfrak{g}$ of a reductive Lie group $G$, then the quotient $\mathfrak{g}^*/G$ is isomorphic to $\mathfrak{h}^*/W$ where $W$ is the Weyl group of $G$, and moreover there exists an open subset $\mathfrak{g}_{\mathrm{reg}}^* \subset \mathfrak{g}^*$ such that its complement in $\mathfrak{g}^*$ has codimension 2 and that the natural map $\mathfrak{g}_{\mathrm{reg}}^* \to \mathfrak{h}^*/W$ is smooth, see [58, Section 3.1]. Then it follows from the aforementioned facts that $\mathfrak{g}^* \times_{\mathfrak{h}^*/W} \mathfrak{h}^*$ is Cohen-Macaulay, and it contains a smooth open subset $\mathfrak{g}_{\mathrm{reg}}^* \times_{\mathfrak{h}^*/W} \mathfrak{h}^*$ whose complement in $\mathfrak{g}^* \times_{\mathfrak{h}^*/W} \mathfrak{h}^*$ has codimension 2, thus $\mathfrak{g}^* \times_{\mathfrak{h}^*/W} \mathfrak{h}^*$ is normal [59, Theorem 39]. Since the projection $\mathfrak{g}^* \times_{\mathfrak{h}^*/W} \mathfrak{h}^* \to \mathfrak{g}^*$ is finite and surjective, we see that $\dim \mathfrak{g}^* \times_{\mathfrak{h}^*/W} \mathfrak{h}^* = \dim \mathfrak{g}^*$.

whereas, according to Lemma 5.2:

$$C(X,u)|\lambda\rangle = \prod_{k=1}^{n}\left(u - (n-k)\hbar - \left(\lambda_k - \frac{n-2k+1}{2}\right)\hbar\right)|\lambda\rangle. \tag{5.19}$$

We recognize $\frac{n-2k+1}{2}$ as the $k$-th component of the Weyl vector $\rho$. Comparing the above two equations give us the highest weight (5.17). The Dynkin labels of the weight $\lambda = (\lambda_1, \cdots, \lambda_n)$ are $\lambda_i - \lambda_{i+1} = \hbar^{-1}(r_n - r_{n+1}) = \varrho_i^{\mathbb{C}}$ for $1 \le i \le n-1$. $\qquad\square$

The brane construction of the T-operator involves $n$ D5, $n$ NS5, and $n$ D3 branes – one D3 brane suspended between each pair of five-branes. This setup was also used in [46] to create Wilson lines valued in Verma modules for 4d $GL_{m|n}$ CS theory with highest weights given by locations of NS5 branes in the holomorphic direction shifted by the Weyl vector.[20] It was shown in [60] that the action of quantized Coulomb branch algebras on their Verma modules can be interpreted in terms of monopole operators acting on vortex configurations.

## 5.2 Example: The Q-Operators (Minuscule 't Hooft Lines)

One class of important examples of line operators in 4d $GL_n$ Chern-Simons theory are the Q-operators, also known as the minuscule 't Hooft operators [22]. They are labeled by $Q_0, Q_1, \cdots, Q_{n-1}$ where $Q_k$ in our notation is $\mathbb{T}_\varrho(K, L)$ for the tuples:

$$K = (\overbrace{0, \cdots, 0}^{n-k}, \overbrace{1, \cdots, 1}^{k}), \qquad L = (k). \tag{5.20}$$

$Q_0$ is the trivial line operator whose phase space is just a point, and for $0 < k \le n/2$, the phase space of $Q_k$ is the Coulomb branch of the 3d $\mathcal{N} = 4$ theory of the quiver Fig. 14a. It is known

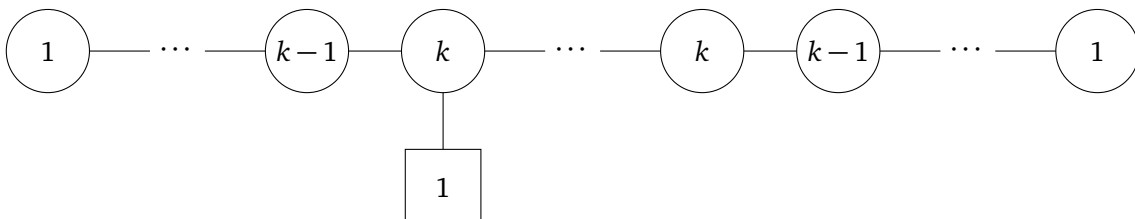

(a) Quiver for the theory $T_\varrho[U(n)]_K^L$ where $K$, $L$ are given by (5.20). There are $n-1$ circles in the quiver. The Coulomb branch of this theory is the phase space of the Q-operator $Q_k$.

(b) The Coulomb branch can also be described as the bow variety for this bow diagram with $n$ crosses.

Figure 14: 3d $\mathcal{N} = 4$ quiver and bow diagram associated with the Q-operator $Q_k$.

that the Coulomb branch of this quiver is the affine space $\mathbb{A}^{2k(n-k)}$ [11], and it quantizes to the Weyl algebra

$$\text{Weyl}_\hbar^{\otimes k(n-k)} = \mathbb{C}\langle x_{ij}, y_{lm} \mid 1 \le i, l \le k < j, m \le n\rangle/([x_{ij}, y_{lm}] = \hbar\delta_{il}\delta_{jm}). \tag{5.21}$$

---

[20]The choice of Weyl ordering (5.8) in the quantum moment map (5.9) and the definition of the generators of $\mathfrak{gl}_n$ (5.12) is important to get the shift by the Weyl vector. It is not clear to us if there is some canonical/physical reason that singles out the Weyl ordering over other ordering schemes.

There is only one mass parameter $\varrho = (z)$ which can be adjoined as a formal variable or evaluated at some complex value to get the algebra (5.21). The phase space can be equivalently described by the bow variety of Fig. 14b.

All the Q-operators are created using a single NS5 brane. The operator $Q_k$ is created by connecting $k$ D5 branes to the only NS5 brane using $k$ D3 branes.

Phase spaces of Q-operators were computed in [22] by directly solving the equations of motion of 4d CS theory in the presence of minuscule 't Hooft lines. The authors further described the quantization of 't Hooft operator in terms of quantization of 3d $\mathcal{N} = 4$ Coulomb branches. The $n$ distinct choices of minuscle coweights of $\mathfrak{gl}_n$ correspond in our example to the choice of $k$.

**Remark 5.4** (All the Q-operators from [21]). In the work of Bazhanov *et. al.* [21], the Q operators are labeled by subsets of $\{1, \cdots, n\}$. The operators $Q_k$ in our paper are denoted by $Q_{\{1,\cdots,k\}}$ in [21], and other Q operators $Q_I$ with $|I| = k$ are obtained by permutation of coordinates. From the phase space perspective, the phase space of $Q_k$ is identified with the cotangent bundle of a big cell of $\mathrm{Gr}(k, n)$ [22], and a matrix $g \in \mathrm{GL}_n$ transforms this phase space to the cotangent bundle of other open cells. If $g$ is taken to be permutation matrix, then we get the Weyl conjugations of the standard big cell, and in total there are $\binom{n}{k}$ such Weyl conjugations, which are in one-to-one correspondence with the operators $Q_I$ with $|I| = k$ in [21].

## 5.3 Example: The L-Operators (Wilson-'t Hooft)

A class of more complicated examples of line operators in 4d Chern-Simons theory are L-operators. They are labeled by $L_0, L_1, \cdots, L_n$, where $L_0$ is the trivial line operator whose phase space is just a point, $L_n$ is the T-operator from §5.1, and for $0 < k < n$, the phase space of $L_k$ is the Coulomb branch of the 3d $\mathcal{N} = 4$ theory of the quiver Fig. 15a. We claim that the

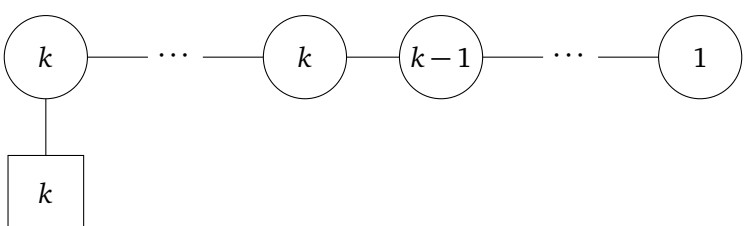

(a) Quiver for the 3d $\mathcal{N} = 4$ theory whose Coulomb branch corresponds to the phase space of the L-operator $L_k$. There are $n-1$ circles in the quiver. The Coulomb branch quantizes to the algebra (5.22)

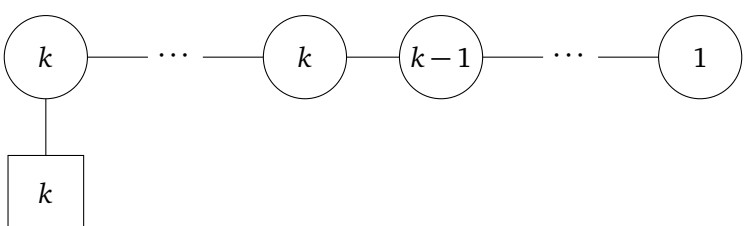

(b) Bow diagram with $n$ crosses whose associated variety is the Coulomb branch of the above 3d $\mathcal{N} = 4$ quiver.

Figure 15: 3d $\mathcal{N} = 4$ quiver and bow diagram associated with the L-operator $L_k$.

quantized Coulomb branch algebra of this quiver is

$$U_\hbar(\mathfrak{gl}_k) \otimes \mathrm{Weyl}_\hbar^{\otimes k(n-k)}. \tag{5.22}$$

In fact, using [56, Lemma 6.3] we see that the Coulomb branch algebra of this quiver is the $\mathrm{GL}_k$ invariant subalgebra of $\mathrm{Weyl}_\hbar^{\otimes kn}$, where $\mathrm{GL}_k$ embeds into $\mathrm{GL}_n$ diagonally and acts on the second tensor component of $\mathbb{C}^{kn} = \mathbb{C}^k \otimes \mathbb{C}^n$. Thus, the Coulomb branch algebra is

$(\mathrm{Weyl}_{\hbar}^{\otimes k^2})^{\mathrm{GL}_k} \otimes \mathrm{Weyl}_{\hbar}^{\otimes k(n-k)}$, and it is easy to see that $(\mathrm{Weyl}_{\hbar}^{\otimes k^2})^{\mathrm{GL}_k} \cong U_{\hbar}(\mathfrak{gl}_k)$. The phase space of the L-operator can be equivalently described by the bow variety associated to the diagram Fig. 15b.

In 4d CS theory the L-operator is a dyonic line whose electric charge corresponds to a representation of $U_{\hbar}(\mathfrak{gl}_k)$ determined by mass parameters $(\varrho_1^{\mathbb{C}}, \cdots, \varrho_{k-1}^{\mathbb{C}})$ and a spectral parameter $z$ that are related to the $r_i$'s from Fig. 15b by a relation similar to (5.5). The magnetic charge is determined by a minuscule coweight of $\mathfrak{gl}_n$ (cf. §5.2).

## 5.4 Open Bow Diagrams

We define an open bow diagram to be a linear diagram with circles and crosses such that it is allowed to have nothing on one or both ends, i.e. there could be semi-infinite D3 branes which do not end on five-branes. And we define the open bow variety associated to an open bow diagram to be similar to a bow variety, except that the symmetries at the open ends are not being gauged or quotient out. For example, Fig. 16 is an open bow diagram whose corresponding open bow variety is well-known to be $T^*\mathrm{GL}_k$. It is a D5 type boundary providing

$$\overset{\mathrm{GL}_k}{\underset{\times}{\sim\!\!\sim}} \overset{k-1}{\underset{\cdots}{\sim\!\!\sim}} \overset{2}{\underset{\times}{\sim\!\!\sim}} \overset{1}{\underset{\times}{\sim\!\!\sim}} \times$$

Figure 16: The open bow variety corresponding to pure Dirichlet boundary condition. The $\mathrm{GL}_k$ over the open edge, as opposed to just $k$, is meant to imply that the corresponding $\mathrm{GL}_k$ symmetry has not been gauged.

the pure Dirichlet boundary condition.

Note that for any Poisson variety $\mathcal{X}$ with a Hamiltonian $\mathrm{GL}_k$ action, the Hamiltonian reduction $(\mathcal{X} \times T^*\mathrm{GL}_k) /\!\!/ \mathrm{GL}_k$ is isomorphic to $\mathcal{X}$ as a Poisson variety. This implies that gluing the above open bow diagram to some other open bow diagram does not change the variety, therefore we see that an open bow variety of a given open bow diagram is actually isomorphic to the bow variety associated to the new bow diagram formed by gluing the open ends of the above bow diagram and the given bow diagram. In terms of branes, we then get an open bow variety by allowing the D3 branes to end on D5 branes such that all of these D5 branes acquire linking number 1.

The identification between open bow variety and bow variety also holds at the quantum level. In fact, it is known that $T^*\mathrm{GL}_k$ quantizes to the ring $D_{\hbar}(\mathrm{GL}_k)$ of differential operators on $\mathrm{GL}_k$, and for any $\mathbb{C}[\hbar]$-algebra $\mathcal{A}$ equipped with a Hamiltonian $\mathrm{GL}_k$-action, the quantum Hamiltonian reduction $(\mathcal{A} \otimes D_{\hbar}(\mathrm{GL}_k)) /\!\!/ \mathfrak{gl}_k$ is isomorphic to $\mathcal{A}$.

We can glue any two open bow varieties that have the same ranks at their open edges. Let $\mathcal{A}_L$ and $\mathcal{A}_R$ be two algebras associated to two open bow diagrams with rank $k$ at their open edges. The same rank $k$ at the open edges means both $\mathcal{A}_L$ and $\mathcal{A}_R$ are equipped with Hamiltonian $\mathrm{GL}_k$ actions. Then the glued variety quantizes to the quantum Hamiltonian reduction $(\mathcal{A}_L \otimes \mathcal{A}_R) /\!\!/ \mathfrak{gl}_k$. This leads to the notion of boundary algebras, such that we get the algebra associated to a bow variety by gluing two boundary algebras. Similar boundary algebras were studied by Dedushenko-Gaiotto [45,53] where they not only constructed boundary algebras but also their traces by computing correlation functions in a hemisphere background (as opposed to our $\Omega$-background).

In terms of branes, the gluing corresponds to identifying D5 branes at the Dirichlet boundaries, joining D3s from opposite sides of the same D5, and removing the D5s from the configuration. We illustrate the process for $k = 3$ in Fig. 17.

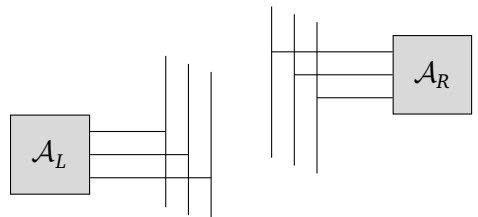

(a) Two different brane configurations with Dirichlet boundary conditions at one end, corresponding to the algebras $\mathcal{A}_L$ and $\mathcal{A}_R$.

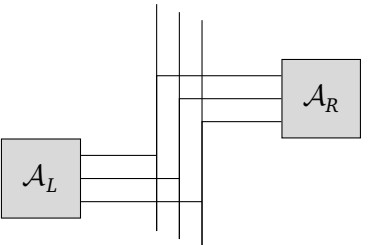

(b) Identifying the D5 branes from the two different Dirichlet boundaries.

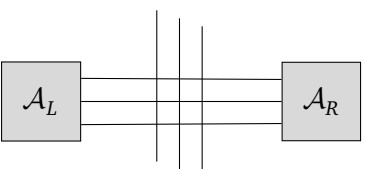

(c) Joining the D3 branes from opposite sides of the D5 branes.

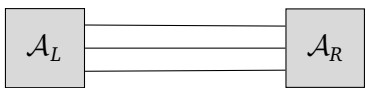

(d) The D5 branes are moved away and we are left with the configuration corresponding to the algebra $(\mathcal{A}_L \otimes \mathcal{A}_R) /\!\!/ \mathfrak{gl}_3$.

Figure 17: Gluing two boundary algebras $\mathcal{A}_L$ and $\mathcal{A}_R$ with Hamiltonian GL$_3$ actions – the brane picture. The gray squares with $\mathcal{A}_L$ and $\mathcal{A}_R$ hide whatever configurations of branes are needed to get the algebras $\mathcal{A}_L$ and $\mathcal{A}_R$. The vertical and horizontal lines are D5 and D3 branes respectively.

# 6 Conclusion

Let us end with a few remarks on open threads. Instead of putting the line operators $\mathbb{L}_\varrho^z(K, L)$ horizontally just to create a monodromy operator as in Fig. 11, we can put a collection of these operators vertically to create the spin chain itself. We then get direct products of bow varieties as phase spaces of these spin chains.[21] It will be interesting to consider geometric and brane quantization of these phase spaces following [61–63], especially considering that quantization of these bow varieties should result in quantum integrable systems according to this construction.We are further exploring operator relations in 4d CS theory corresponding to the famous TQ and QQ-relations [64–67] and their relations to fusions of bow varieties. When the phase spaces can be identified with vacuum moduli spaces of 3d $\mathcal{N} = 4$ quiver gauge theories, these results are along the lines of many known relations between integrable spin chains and such 3d theories [15, 16, 68–72]. Known connections between spin chains and 3d gauge theories with Hanany-Witten type brane construction was used in [69] to establish bispectral dualities of integrable spin chains. Brane constructions similar to that of §5.1 were used in [46] to create spin chains with Verma modules and to illustrate fermionic dualities of superspin chains. We hope that our brane constructions and characterization of line operators using Cherkis bow varieties will further aid the study of non-trivial dualities of integrable spin chains and related gauge theories.

# Acknowledgments

**Funding information**    Majority of the work done by NI for this paper took place at the Institute for Advanced Study in Princeton, NJ 08540, USA with support from the National Sci-

---

[21]Assuming that the 4d CS theory itself has a trivial phase space (a single point) without the lines.

ence Foundation under Grant No. PHY-1911298. At IHÉS NI is supported by the Huawei Young Talents Program Fellowship. Kavli IPMU is supported by World Premier International Research Center Initiative (WPI), MEXT, Japan. YZ also thanks Perimeter Institute for Theoretical Physics in Waterloo, ON N2L 2Y5, Canada, where part of his work was done as a graduate student.

## A  $\Omega$-deformed Supersymmetry

Let $V$ be a U(1) space-time symmetry acting on $\mathbb{R}^2_{12}$ – the world-volume of the 2d B-model from §4.1. Turning on $\Omega$-deformation using $V$ deforms the B-model supersymmetry (4.14), (4.19), and (4.23). We denote the deformed supersymmetry variation by $\delta_V$. It acts on the B-model multiplets as follows.

On vector multiplet (deformation of (4.14)):

$$
\begin{aligned}
\delta_V \mathcal{A}^{(2)} &= \iota_V \chi^{(2)}, & \delta_V \overline{\mathcal{A}}^{(2)} &= \psi^{(2)} - \iota_V \chi^{(2)}, \\
\delta_V \psi^{(2)} &= 2\iota_V F^{(2)} - 2\mathrm{i} D^{(2)} \iota_V \phi^{(2)}, & \delta_V \eta &= P, \\
\delta_V P &= \iota_V \mathcal{D}^{(2)} \eta, & \delta_V \chi^{(2)} &= \mathcal{F}^{(2)}.
\end{aligned}
\tag{A.1}
$$

On the first chiral multiplet (deformation of (4.19)):

$$
\begin{aligned}
\delta_V \mathcal{A}^{(c)} &= \iota_V \chi^{(c)}, & \delta_V \overline{\mathcal{A}}^{(c)} &= \psi^{(c)} - \iota_V \chi^{(c)}, \\
\delta_V \chi^{(c)} &= \mathrm{d}^{(c)} \mathcal{A}^{(2)} + \mathcal{D}^{(2)} \mathcal{A}^{(c)} + \iota_V \mathsf{F}, & \delta_V \left( \psi^{(c)} - \iota_V \chi^{(c)} \right) &= \iota_V \left( \mathrm{d}^{(c)} \mathcal{A}^{(2)} + \mathcal{D}^{(2)} \overline{\mathcal{A}}^{(c)} \right), \\
\delta_V \mathsf{F} &= \mathcal{D}^{(2)} \chi^{(c)} + \mathcal{D}^{(c)} \chi^{(2)} & \delta_V \overline{\mathsf{M}} &= \overline{\mathsf{F}}, \\
& & \delta_V \overline{\mathsf{F}} &= \mathcal{D}^{(2)} \iota_V \overline{\mathsf{M}}.
\end{aligned}
\tag{A.2}
$$

On the second chiral multiplet (deformation of (4.23)):

$$
\begin{aligned}
\delta_V \sigma &= \iota_V \overline{\psi}^{(2)}, & \delta_V \overline{\sigma} &= \overline{\eta}, \\
\delta_V \overline{\psi}^{(c)} &= \mathcal{D}^{(2)} \sigma + \iota_V \mathsf{G}, & \delta_V \overline{\eta} &= \iota_V \mathcal{D}^{(2)} \overline{\sigma}, \\
\delta_V \mathsf{G} &= \mathcal{D}^{(2)} \overline{\psi}^{(c)} + \sigma \wedge \chi^{(2)}, & \delta_V \overline{\mathsf{N}} &= \overline{\mathsf{G}}, \\
& & \delta_V \overline{\mathsf{G}} &= \mathcal{D}^{(2)} \iota_V \overline{\mathsf{N}}.
\end{aligned}
\tag{A.3}
$$

The deformed supersymmetry satisfies the algebra (deformation of (4.24)):

$$
\delta_V^2 = \mathcal{D}^{(2)} \iota_V + \iota_V \mathcal{D}^{(2)} = \mathcal{L}_V + \delta^{\text{gauge}}_{\iota_V \mathcal{A}^{(2)}}.
\tag{A.4}
$$

Here $\mathcal{L}_V$ is Lie derivative with respect to $V$ and $\delta^{\text{gauge}}_{\iota_V \mathcal{A}^{(2)}}$ denotes gauge transformation generated by $\iota_V \mathcal{A}^{(2)}$. The $\Omega$-deformed theory localizes to the fixed point of $V$ with $\mathcal{L}_V$-invariant fields and action given by the $\mathcal{L}_V$-invariant superpotential of the B-model.

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
