# Peer review of "Line Operators in 4d Chern-Simons Theory and Cherkis Bows"

_SciPost Physics, doi:SciPost Phys. 16, 052 (2024)_

## Round 2 · Referee Report · Anonymous (Referee 1) · 2023-4-14

Report

This work overall is quite interesting and I enjoyed reading it.
At the same time, while reading it, I could not shake off the feeling that the paper is merely a compilation of a bunch of known facts, with little real novelty.

Nevertheless, in the end, it seems to contain some interesting material overall. Therefore, it probably should be published, and it likely contains something new.
Let me, however, first try to separate things that can be separated by asking a few questions:

(1) Is 4d CS really crucial for the connection to bow varieties? Seems like not really. Correct me if I'm wrong. The bow variety describes the Higgs branch of a 3d N=4 theory arising from the brane construction of a codim-3 defect in the 6d (1,1) SYM. Then the fact the upon Omega-deformation, this 6d-3d SUSY system reduces to a 4d-1d coupled system (4d CS with a line defect), is a completely independent fact. Correct?

(2) If (1) is true, then can you identify which part of the construction is really novel? Also, what is the connection to reference [54], and can one do anything like that to the general line operators you build? Maybe the 4d CS provides some unique perspective on these lines that goes beyond the brane description of a 3d N=4 theory? If that is the case, it would be interesting and new.

Additionally, upon carefully reading the draft, I gathered a collection of questions, suggestions, concerns, and typos, as listed below.

Requested changes

  1. The paper starts by talking about phase spaces of line operators, but never defines what it is. While the phase space of the entire system is a familiar object, it would help to say a word or two on what is meant by the phase space of a line operator (otherwise a reader has to read the paper first to extract the definition).

  2. Top of page 3: "This fits nicely with the fact that the line operators in 4d CS theory form integrable spin chains which carry natural Yangian actions" -- it is, actually, not entirely clear what fits with this fact and why.

  3. Page 3, second paragraph, the reference to [20] and the statement that those authors construct L operators via monodromy lines: monodromy lines in which theory? Then there's a sentence mentioning monodromy lines for the Q operators and lines for the T operators: again, what sort of lines are these? In which theory? Authors of [20] most definitely didn't work with the 4d CS, hence it is not clear what's the context for lines mentioned in connection to that reference.

  4. Last paragraph of the introduction, second sentence: "the" is repeated twice.

  5. Last paragraph of the introduction, last sentence: "...algebras related to the T, Q, and L-operators..." -- what is meant by an algebra related to the T/Q/L-operator?

  6. After (2.1): $\vec{\omega}\cdot {\rm d}\vec{x}$ is a one-form, hence in ${\rm d}U = \star_{\mathbb{R}^3}(\vec{\omega}\cdot {\rm d}\vec{x})$, the form degrees do not match. In fact, there should be one more $\rm d$ on the right, applied before taking $\star$.

  7. Second paragraph on pg.5: I think the statement that Omega-deformed SYM "reduces" to the 4d CS is really at the level of Q-cohomology, or "protected sector". Just a clarification.

  8. Footnote 2: TQM has zero Hamiltonian, therefore, saying "no additional potential term is present in the Hamiltonian" is misleading. There is simply no Hamiltonian in the first order action, only the symplectic term.

  9. Caption of Figure 1 says that complex FI parameters are not visible in the classical brane picture -- this seems misleading. There is nothing "quantum" about such FI parameters, it is simply hard to depict them on a planar brane diagram, but they have clear geometric meaning.

  10. After (2.3), about dropping the stability condition from the notation: can there be different, inequivalent stability conditions?

  11. Page 7: how reliable is the anaysis that uses 6d (1,1) SYM, given that it's an IR-free theory?

  12. About (2.6): why does it have to be a projection to $F_{\mathbb{C}}$ (which seems to require a choice), rather than the canonically available injection $F_{\mathbb{C}} \to GL_n$? On the one hand, I indeed could project the $gl_n$ valued bulk gauge field onto $f_{\mathbb{C}}$ to define the defect coupling. On the other, I could consider injecting the $f_{\mathbb{C}}$ valued current on the defect into the $gl_n$-valued current in the bulk. What is the reason to prefer the former?

  13. The definition of phase space in (2.7) as a Higgs branch depends crucially on the theory $T^\vee$. Is this notion invariant at all? What if the defect (as a 4d-1d coupled system) has a dual description, with a different theory $T^\vee$, whose Higgs branch is different (but somehow, the defect becomes the same after coupling T^\vee to the bulk)? The definition (2.7) doesn't make it manifest that the phase space defined as the Higgs branch is an invariant. Is there a better definition of the line operator phase space, which makes it manifestly intrinsic and independent of the concrete realization?

  14. On the sentence right after the eqn. (2.7): the TQM itself already quantizes $M_H$, one does not need CS. The TQM-CS dynamics seems to further gauge some isometries of $M_H$. Hence a question: does the $\mathcal{A}_\varrho$ from (2.8) just quantize $M_H$, or does it also include the effects of gauging (some kind of quantum Hamiltonian reduction)?

  15. Equation (2.9) has $v_j$ without tilde, while in the Figure 3, v's appear with tildes. Seems like a typo?

  16. Between (2.11) and (2.12): "...SU(2) symmetry rotating the complex symplectic structures..." -- it rotates the complex structures and it rotates the symplectic structures, but did you really mean to say that it rotates the complex symplectic structures? (well, it does, but maybe you didn't mean precisely that?)

  17. Before eqn. (2.12): symplectic reduction, subject to the stability condition.

  18. Strictly speaking, in (3.2) one has Spin groups.

  19. Right after (3.7): "...the associated current must vanish at that boundary." -- not the whole current, only its normal component.

  20. Section 3.2 starts with an unmotivated assumption that the correct twist must belong to the KW family. Why? The 4d N=4 SYM has a few other inequivalent twists, -- why is it only the KW twist that you study?

  21. Right before Section 4: the statement that one obtains 2d BF theory seems umotivated at this point. Need some justification/references.

  22. The upper indices on dx's in (4.1) should probably be 0,1,2,3?

  23. Remark on the style: between eqns (4.21) and (4.22), it is best to avoid too many "will". Better to say: "We integrate out..... after which we are lead to..."

  24. Typo: after (4.47), there should be no "is" after the word "derivative".

  25. Page 22: "The actual value of m and t are coordinate dependent and physically irrelevant..." -- the statement that physical couplings are physically irrelevant (which, by the way, are relevant couplings in the techincal sense) seems a bit problematic. Please make this statement more precise, reflecting what you really mean.

  26. In the end of paragraph before (4.87), the representation spaces R and R' appear out of nowhere, without any definition. Please clarify. Also, remove doubled "the" in the last sentence before (4.87).

  27. Between (5.1) and (5.2): what does it mean for an algebra to "couple" to a line operator?

  28. Page 33, before section 5.1. "monodromy matrices associated with the algebras U, Weyl, ..." -- what does it mean to associate the monodromy matrix with an algebra?

  29. Proposition 5.1: "where the Dynkin labels of lambda and rho'' should be separated by commas.

  30. Next paragraph "values of the casimirs" is missing "of".

  31. Paragraph before (5.5): "If we turn on all complex twisted masses and treat them as undetermined parameters, then the Coulomb branch is $gl_n^*$" -- provide a reference for this statement.

  32. Statement about the quantum Hamiltonian reduction before (5.6): please also provide a reference.

  33. Paragraph before Remark 5.2: "It is known that ... is a normal variety of dimension n^2, and ... is the ring of function on a ..." -- if it is known, please include the appropriate reference.

  34. Just before (5.17): ".. acts on the Verma module with the highest weight..." -- where does the Verma module come from? It appeared unmotivated, out of nowhere, basically.

  35. End of Section 5.1, right before Section 5.2: "...quantized Coulomb branch algebras of 3d N=4 theories act on Verma modules;..." -- this statement is a tautology. Indeed, any algebra is going to act on its Verma modules, if there are any. Please clarify what you actually meant to say.

  36. Page 37, last paragraph: "Phases spaces" should be "Phase spaces".

  37. Remark 5.4: "The operators $Q_k$ in our paper is denoted" should be "are denoted". Also in "other Q operators $Q_I$ with $|I|=k$ is obtained" -- should be "are obtained".

  38. Right before (5.22): "algebra of the this quiver" -- delete "the".

  39. first paragraph in Section 5.4: "And we define the open bow variety associated to an open bow diagram to be similar to that of a bow variety" -- looks like "that of" is unnecessary.

  40. Later on page 39: "oepn" ---> "open".

  41. Later on page 39: "also hold at the quantum level" should have "holds".

  42. Caption of Figure 17: "..whatever configurations of branes is needed..." -- should have "are needed".

  43. The procedure described on page 40 seems to be simply "gluing by gauging", which is well known in the literature. Is it?

  44. Conclusions: "Brane constructions similar to that of section 5.1 was used in [44]" -- should have "were used".

  45. Provide references for the Appendix A.

  • validity: high
  • significance: good
  • originality: ok
  • clarity: high
  • formatting: perfect
  • grammar: excellent

Author:  Nafiz Ishtiaque  on 2023-04-22  [id 3608]

(in reply to Report 1 on 2023-04-14)
Category:
remark
answer to question
reply to objection
correction
pointer to related literature

We thank the referee for the thorough and careful review. Please see the attached file for our response.

Attachment:

reply_to_report.pdf

---

## Round 3 · Referee Report · Anonymous · 2023-11-23

Report

See the attached pdf file.

Attachment

  • validity: -
  • significance: -
  • originality: -
  • clarity: -
  • formatting: -
  • grammar: -

Author:  Nafiz Ishtiaque  on 2024-01-13  [id 4245]

(in reply to Report 1 on 2023-11-23)

We thank the referee for the careful review. We also thank the referee for bringing three relevant references, especially Cherkis, O'Hara, Saemann, Phys. Rev. D, 83:126009 to our attention. This latter one clearly derives the relation between the bow varieties and supersymmetric vacuum equations of 4d N=4 theories with impurity walls. We presented this relation in our paper as part of our results -- but clearly, the paper by Cherkis, O'Hara, and Saemann is in fact the original reference for this result, of which we were regrettably unaware. We also present the equivalence between the vacuum equations and bow varieties in the context of 2d BF theory, which we hope will be of interest to researchers working with topological twists of supersymmetric gauge theories. This and the other references will be added to the new version of the paper. For more detailed responses to the other points mentioned in the review please see the attached document.

Attachment:

reply_to_report.pdf

---

## Round 3 · List of Changes

Main changes:
1. Clarified the meaning of phase space, second paragraph of section 1.
2. Clarified the meaning of "monodromy lines", 3rd paragraph of page 2.
3. Added a paragraph at the end of section 1 summarizing the new results of the paper.
4. Added a paragraph at the beginning of section 3.2 explaining the reason for considering the Kapustin-Witten twist of 4d N=4 in the given context, as opposed to some other topological twist.
5. Added footnote 18.

Additionally, several typos have been corrected and minor modifications to some sentences have been made to clarify their meaning based on the referee report on the previous submission.

---

## Round 4 · Referee Report · Anonymous · 2024-1-20

Report

I recommend this paper for publication in its current form.

---

## Round 4 · Referee Report · Anonymous · 2024-1-21

Report

After a long refereeing process, the authors have addressed a lot of problems that were present in the original manuscript, and answered many questions asked by the referees. There may still be some minor issues here and there (see, e.g., my comment to the author's reply to the first referee report). However, at this point I can confidently recommend this article for publication in the current form.

---

## Round 4 · List of Changes

Based on the last referee report:
1. New citation added for existing result relating bow equations to the vacuum equations of 4d N=4 with impurity walls (second paragraph, p. 3).
2. References cited for:
a) The result that 2d B-model with superpotential W localizes to 0d theory with action W upon omega deformation, concerning eq. 4.46.
b) Quantization of $gl_n^*$ to the universal enveloping algebra $U_\hbar(gl_n)$ upon turning on twisted masses (paragraph containing eq. 5.5).
c) Localization of 3d N=4 theory to topological quantum mechanics by omega deformation (second paragraph of section 2.1).
3. In the second paragraph on p.2: The statement regarding coupling a quantum mechanics with global symmetry to a gauge theory changed slightly to make it clearer.
4. Support of $\vec \omega$ corrected after eq. 2.1.
5. Mentioned $K_i$ and $L_p$ to be integers, before eq. 2.2.
6. Spelled out the meaning of "unsymmetrization" in Remark 5.3.
7. Corrected several spelling errors.

---

## Editorial Decision

published